# SURE: Semantic Uncertainty Regularization for Test-Time Adaptation in Vision-Language Models

## Abstract

Test-time adaptation (TTA) aims to improve model robustness under distribution shift by exploiting unlabeled test data. Existing methods often rely on pseudo-labels, which are noisy and treated independently, ignoring both their temporal reliability and the semantic structure of the label space. We introduce SURE (Semantic Uncertainty REgularization), a framework that regularizes predictions through a dynamically evolving prototype-reliability graph (PRG). PRG captures semantic affinity across classes and the stability of confidence over time, enabling the selective propagation of reliable predictions while suppressing errors. This structure-driven regularization enforces semantic consistency and prevents error amplification. Across diverse domain-shift benchmarks, SURE consistently outperforms prior methods, offering a principled and generalizable approach to reliable TTA.

## 1 Introduction

Vision-language models (VLMs) such as CLIP (Radford et al., 2021) and ALIGN (Jia et al., 2021) achieve strong zero-shot transfer by aligning images and text in a shared embedding space. Yet, their performance degrades under distribution shifts—*e.g.*, changes in appearance, context, or class priors—resulting in uncertainty and semantic misalignment. Compared to unimodal vision models, this degradation is particularly severe for VLMs, since distribution shifts may not only distort visual features but also disrupt their alignment with textual prototypes. This motivates test-time adaptation (TTA) for robust deployment.

Recent work has explored TTA for VLMs without labeled supervision or source data (Shu et al., 2022; Zanella & Ben Ayed, 2024; Ma et al., 2024; Yoon et al., 2024). Entropy-based methods (Shu et al., 2022; Sui et al., 2024; Zhang et al., 2024a) adapt by minimizing prediction entropy over augmented views, while prototype-based methods (Zanella & Ben Ayed, 2024; Zhou et al., 2025) refine class prototypes guided by text embeddings. Despite different formulations, both families rely heavily on model predictions, making them vulnerable to noisy pseudo-labels under distribution shift. In practice, these approaches suffer from distinct failure modes: entropy minimization can render the model overconfident in incorrect predictions, whereas early prototype updates may propagate noise and destabilize adaptation.

Confidence thresholding used in Shu et al. (2022) partly alleviates noise but discards informative low-confidence samples. We argue that adaptation should instead exploit the evolving reliability of class-level predictions, reflecting the model's accumulated certainty about each category. Rather than treating adaptation as isolated prediction correction, we advocate a structured paradigm where adaptation emerges from the interaction between evolving predictions and semantic priors. To this end, we formulate adaptation as a structured process that propagates reliable information while suppressing semantic noise, laying the foundation for our proposed method.

As shown in Fig. 1, we move beyond instance-level heuristics and propose a structured formulation of test-time adaptation. The key idea is to regularize predictions through an evolving Prototype-Reliability Graph (PRG), where edges encode (i) semantic affinity derived from textual prototypes and (ii) class-wise reliability measured by the temporal stability of pseudo-label confidences. PRG acts as a semantic regularizer: it propagates support from confident and semantically related classes

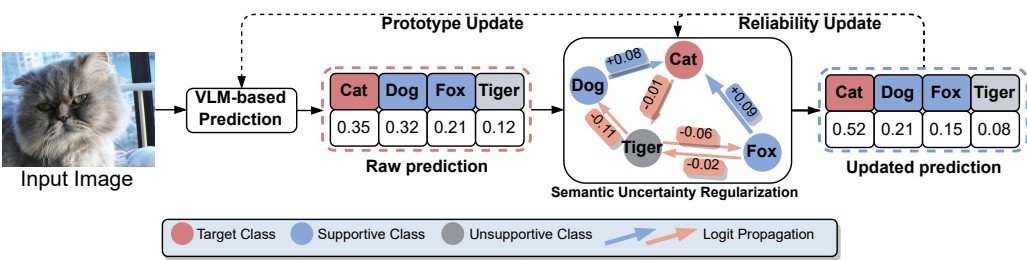

Figure 1: Conceptual diagram of SURE. At test time, raw predictions are refined through a prototype-reliability graph, where edges encode both semantic similarity and class-wise reliability. As the input stream arrives, the graph is dynamically updated via prototype and reliability updates, aligning predictions with evolving semantic priors. This structured propagation emphasizes support from semantically related and statistically stable neighbors while suppressing noisy signals, thereby enforcing semantic consistency and mitigating error amplification under distribution shift.

to uncertain ones, while suppressing unreliable associations. Concretely, logits are updated via graph-based propagation, aligning outputs with structure-induced consistency constraints. We instantiate this principle as SURE (Semantic Uncertainty REgularization), a closed-loop framework in which predictions, prototypes, and graph structure co-evolve, thus stabilizing adaptation and mitigating semantic drift under distribution shift.

- We propose SURE, a test-time adaptation framework that regularizes prediction dynamics via a Prototype-Reliability Graph, jointly modeling semantic affinity and class-wise reliability.

- We design a closed-loop adaptation mechanism where predictions and graph structure co-evolve, realized through prototype refinement, reliability update, and graph-based logit propagation.

- We demonstrate the effectiveness and generality of SURE across four domain-shift benchmarks and two VLM backbones, achieving consistent robustness gains and state-of-the-art performance.

## 2 RELATED WORK

**Adapting vision-language models.** Vision-language models such as CLIP (Radford et al., 2021), ALIGN (Jia et al., 2021), and BLIP (Li et al., 2022) achieve strong zero-shot transfer through large-scale contrastive pretraining. To further improve downstream performance, parameter-efficient adaptation methods have been proposed. Prompt-learning approaches such as CoOp (Zhou et al., 2022b) and CoCoOp (Zhou et al., 2022a) optimize soft prompts, while adapter-based methods (Zhang et al., 2022b; 2023) insert lightweight modules into frozen backbones. Structure-aware adapters like GraphAdapter (Li & Jiang, 2025) further introduce graph neural networks to model class correlations within the adapter framework, enhancing the semantic consistency of learned features. Other variants such as LoRA (Hu et al., 2022) and fine-tuning extensions have also been explored. While effective, these approaches typically require labeled supervision and offline training, making them unsuitable for deployment-time adaptation under distribution shift.

**General-purpose test-time adaptation.** Standard TTA aims to adapt a pre-trained model to unlabeled test streams via self-supervised objectives. Pioneering works like TENT (Wang et al., 2020) update batch normalization parameters by minimizing prediction entropy. Schneider et al. (Schneider et al., 2020b) demonstrated that simply replacing training-time batch statistics with test-time statistics significantly improves robustness against corruptions. T3A (Iwasawa & Matsuo, 2021) adjusts the classifier prototype using pseudo-labeled samples, while SAR (Niu et al., 2022) introduces sharpness-aware minimization to select reliable samples. Other approaches focus on distribution alignment (Liang et al., 2020), conjugate pseudo-labels (Goyal et al., 2022), or robustifying normalization statistics (Zhang et al., 2022a; Schneider et al., 2020a). However, these methods generally operate on uni-modal (vision-only) architectures. Directly applying them to VLMs is often suboptimal because they fail to exploit the rich semantic prior encapsulated in the pre-trained text encoder.

**Test-time adaptation of VLMs.** TTA offers a label-free paradigm for adapting VLMs at inference time using only unlabeled inputs. Existing methods can be broadly grouped into: (i) *entropy-based* approaches (Shu et al., 2022; Sui et al., 2024; Imam et al., 2025; Yoon et al., 2024; Sheng et al., 2025) that minimize prediction entropy to calibrate features; (ii) *prototype-based* approaches (Zanella & Ben Ayed, 2024; Zhang et al., 2024a;b; Zhou et al., 2025) that build or refine prototypes guided by text embeddings; (iii) *ensemble and augmentation* methods such as ZERO (Farina et al., 2024); and (iv) *optimization-based extensions* including reward-driven (Zhao et al., 2023), MAP estimation (Fuchs et al., 2025), and retrieval augmentation (Lee et al., 2025). Most of these methods treat classes independently and rely on per-instance confidence, overlooking inter-class structure. While we also target VLM adaptation, we diverge from these works by explicitly modeling the topological relationships between classes to rectify unreliable predictions.

**Graph-based reasoning and uncertainty modeling.** Graph-based reasoning captures structured dependencies across entities for tasks like scene understanding and relational prediction (Kipf, 2016; Veličković et al., 2017). In the context of adaptation, PROGRAM (Sun et al., 2024) recently proposed a prototype graph model to propagate pseudo-labels between prototypes and test samples, utilizing message passing to improve pseudo-label quality. While sharing the graph-based spirit, our approach differs from PROGRAM in two key aspects: (1) **Reliability-driven topology**: Instead of a static or purely feature-distance-based graph, we dynamically modulate edges using class-wise uncertainty statistics (reliability), ensuring that noise does not propagate through the graph. (2) **VLM-specific design**: PROGRAM is a general TTA method designed for uni-modal classifiers, whereas our framework is tailored for VLMs, leveraging the frozen text encoder to initialize semantically meaningful graphs. More recently, uncertainty has been integrated into graph learning to yield calibrated inference (Ni et al., 2025; Huang et al., 2023; Han et al., 2025). Unlike these efforts, our approach dynamically constructs a class-level graph during test-time adaptation, coupling semantic similarity with reliability estimation to regularize predictions in a lightweight and principled manner.

## 3 PRELIMINARIES

**CLIP.** CLIP (Radford et al., 2021) is a vision-language model pretrained on large-scale image-text pairs via a contrastive objective that aligns visual and textual embeddings. It consists of an image encoder $E_I(\cdot)$ and a text encoder $E_T(\cdot)$. Given an input image $\mathbf{x}$, the image feature is $\mathbf{f} = E_I(\mathbf{x})$. Textual prompts such as "a photo of a {class name}" are encoded into class prototypes $\mathbf{T} = \{\mathbf{t}_i\}_{i=1}^{C}$, where $\mathbf{t}_i = E_T(\text{"a photo of a {class name}"})$ and $C$ is the number of classes. Classification is then performed via cosine similarity followed by a softmax:

$$p(y_i|\mathbf{x}) = \frac{\exp(\cos(\mathbf{f}, \mathbf{t}_i)/\tau)}{\sum_{j=1}^{C} \exp(\cos(\mathbf{f}, \mathbf{t}_j)/\tau)}, \tag{1}$$

where $\cos(\mathbf{f}, \mathbf{t}_i) = \frac{\mathbf{f}^\top \mathbf{t}_i}{\|\mathbf{f}\| \|\mathbf{t}_i\|}$ denotes cosine similarity, and $\tau > 0$ is a temperature.

**Likelihood Adaptation.** A prominent line of TTA methods adapts CLIP by updating class prototypes according to the test distribution, thereby refining the likelihood $p(y_i|\mathbf{x})$. Zhou et al. (2025) identify such likelihood-level adaptation as a key mechanism. Specifically, when the predicted class $y_i$ has confidence $p(y_i|\mathbf{x})$ above a threshold $\theta$, its prototype $\mathbf{t}_i$ is updated using a normalized running average with the image feature $\mathbf{f}$. A counter $N_i$ tracks the number of updates per class:

$$\mathbf{t}_i \leftarrow \text{Norm}\left(\frac{N_i \cdot \mathbf{t}_i + \mathbf{f}}{N_i + 1}\right), \quad N_i \leftarrow N_i + 1, \tag{2}$$

where $\text{Norm}(\cdot)$ denotes L2 normalization to keep prototypes on the unit sphere. This continual refinement gradually aligns prototypes with the evolving test distribution, improving zero-shot robustness. However, it also inherits a critical limitation: early updates based on noisy pseudo-labels may accumulate errors, motivating the need for more structured reliability modeling.

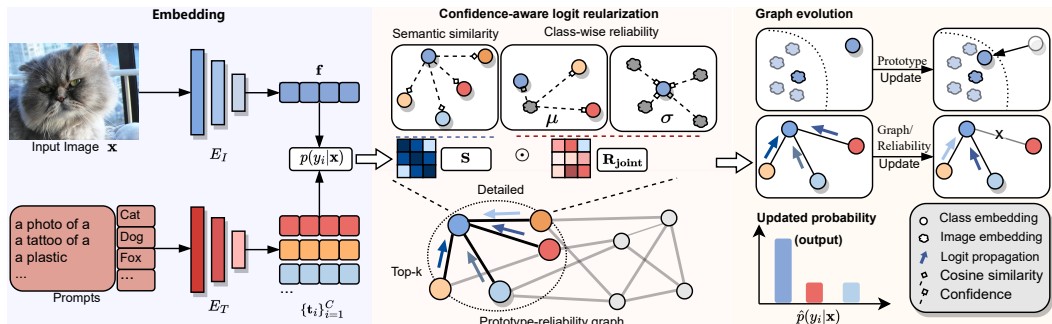

Figure 2: Overview of the proposed test-time adaptation framework SURE. The process comprises three stages: (1) Embedding: Given a test image $\mathbf{x}$, we use a frozen CLIP model to extract its visual embedding and compute initial predictions via cosine similarity to class-specific textual prototypes. (2) Confidence-aware logit regularization: A cached PRG is retrieved, constructed from a semantic similarity matrix $\mathbf{S}$ (computed from textual embeddings) and a joint reliability matrix $\mathbf{R}_{\text{joint}}$ (derived from historical pseudo-label confidence statistics). The PRG is then used to regularize logits and output updated predictions. (3) Graph evolution: Based on updated predictions, we update class prototypes, reliability estimates, and the PRG structure itself via temporal smoothing. This forms a closed-loop semantic regularization process under distribution shift. .

## 4 METHOD

### 4.1 PROTOTYPE-RELIABILITY GRAPH

We propose the Prototype-Reliability Graph (PRG), a dynamically constructed structure that captures robust inter-class relations under distribution shift. Instead of assuming static or uniform similarity, PRG incorporates both semantic affinity and uncertainty-aware reliability to model a class-level graph that reflects the evolving confidence landscape of the target domain.

At each test-time step $\ell$, PRG is instantiated based on the current class prototypes $\{\mathbf{t}_i\}_{i=1}^C$ and pseudo-label confidence statistics. Formally, it can be viewed as a semantic graph whose adjacency structure is modulated by information-theoretic Shannon (1948); Huang et al. (2023) estimates of prediction uncertainty, thereby regularizing connections in favor of reliable, low-entropy class associations. Crucially, unlike standard self-training methods that trust instantaneous high-confidence predictions—which are prone to calibration errors under shift—SURE leverages the temporal statistics of predictions. By computing the variance over a sliding window, we filter out classes that oscillate in confidence, ensuring that only statistically stable predictions guide the adaptation.

**Prototype similarity matrix S.** We compute a semantic similarity matrix $\mathbf{S} \in \mathbb{R}^{C \times C}$ that encodes pairwise affinities between class prototypes:

$$\mathbf{S}_{jk} = \cos(\mathbf{t}_j, \mathbf{t}_k), \tag{3}$$

where $\cos(\cdot, \cdot)$ denotes cosine similarity. In practice, we set $\mathbf{S}_{jj} = 0$ to avoid trivial self-loops. This matrix captures semantic proximity in the embedding space, but remains sensitive to noise when prototypes are distorted by incorrect pseudo-labels under distribution shift.

**Class-wise reliability scores** $R_j$. We modulate the similarity graph using a class-wise reliability score $R_j$, which quantifies the prediction stability associated with class $y_j$. Specifically, we maintain two descriptive statistics in a cache: the mean pseudo-label confidence $\mu_j \in [0, 1]$ and its standard deviation $\sigma_j$, which jointly characterize the historical uncertainty of predictions assigned to class $y_j$. These statistics are updated online as new pseudo-labels arrive. At the beginning of test-time adaptation, we initialize $\mu_j = 1.0$ and $\sigma_j = 0.0$ for all classes to reflect maximal confidence and no variance. The reliability score $R_j$ is then computed as:

$$R_j = \mu_j \cdot \left(1 - \frac{\sigma_j}{\sigma_{\max}}\right), \tag{4}$$

where $\sigma_{\max}$ is a fixed hyperparameter constant (set to 0.5 in our experiments) to ensure stable normalization, independent of batch statistics. In implementation, $R_j$ is clipped to $[0, 1]$ for numerical

stability. This formulation reflects an information-theoretic intuition: reliable classes exhibit low-entropy confidence distributions (high $\mu_j$, low $\sigma_j$), while noisy or ambiguous classes are associated with high entropy. Although $R_j$ is not a direct entropy measure, it serves as a practical proxy for inverse uncertainty and thus quantifies each class's susceptibility to semantic drift.

We next integrate $\mathbf{S}$ and $\{R_j\}$ into a unified adjacency structure that governs graph-based propagation. Specifically, we construct $\mathbf{R}_{\text{joint}}$ and then $\mathbf{W}$ as follows.

**Joint reliability matrix $\mathbf{R}_{\text{joint}}$.** To suppress unreliable semantic interactions, we compute a joint reliability score between class pairs:

$$\mathbf{R}_{\text{joint},jk} = R_j \cdot R_k. \tag{5}$$

This multiplicative form ensures that the trustworthiness of an edge depends on both endpoints: if either class is unstable, the resulting connection is weakened. In this way, $\mathbf{R}_{\text{joint}}$ acts as an edge-wise gating mechanism that penalizes unreliable prototypes and prevents fragile classes from dominating the graph. This design provides the basis for integrating reliability into the subsequent semantic similarity structure.

**Reliability-weighted matrix $\mathbf{W}$.** We then construct a reliability-aware similarity matrix by combining semantic affinity with joint reliability:

$$\mathbf{W} = \mathbf{S} \odot \mathbf{R}_{\text{joint}}, \quad \mathbf{W}_{jj} = 0, \tag{6}$$

where $\odot$ denotes element-wise multiplication and self-connections are set to zero. This formulation downweights unreliable or ambiguous prototype associations, ensuring that unstable classes cannot reinforce themselves. Intuitively, even if a corrupted prototype $\mathbf{t}_j$ appears close to $\mathbf{t}_k$ due to semantic drift, a low reliability score $R_j$ suppresses the edge $\mathbf{W}_{jk}$. The resulting $\mathbf{W}$ defines a reliability-weighted semantic graph, where edges reflect both semantic proximity and prediction stability.

**Construction of sparse adjacency matrix $\mathbf{A}$.** To obtain a sparse and tractable PRG, we identify the top-$k$ neighbors $\mathcal{T}_j^k$ for each class $y_j$ based on the reliability-weighted similarities in $\mathbf{W}$. We retain only these top-$k$ connections and normalize them to define the directed adjacency matrix at test step $\ell$:

$$\mathbf{A}_{jk}^{(\ell)} = \begin{cases} \dfrac{\mathbf{W}_{jk}}{\sum\limits_{y_{k'} \in \mathcal{T}_j^k} \mathbf{W}_{jk'}}, & y_k \in \mathcal{T}_j^k, \\ 0, & \text{otherwise}, \end{cases} \tag{7}$$

yielding $\mathbf{A}^{(\ell)} \in \mathbb{R}^{C \times C}$ as a sparse, directed, and weighted graph. This structure fuses semantic similarity with class-wise reliability and forms the structural backbone for subsequent propagation and prediction refinement.

**Graph Formalization.** With the structural backbone $\mathbf{A}^{(\ell)}$ established, we formally define our PRG framework as a dynamic system $\mathcal{G}^{(\ell)} = (\mathcal{V}, \mathcal{E}^{(\ell)}, \mathcal{M}, \mathcal{U})$:

- **Nodes $\mathcal{V}$:** The set of class prototypes $\{\mathbf{t}_1, \ldots, \mathbf{t}_C\}$. Each node carries a state vector $\mathbf{h}_i = [\mathbf{t}_i, \mu_i, \sigma_i]$ comprising the semantic prototype and its reliability statistics (Eq. 4).

- **Edges $\mathcal{E}^{(\ell)}$:** The directed connections defined by the sparse adjacency $\mathbf{A}^{(\ell)}$. An edge $(j \to i)$ represents a reliability-weighted semantic dependency derived from Eq. 6.

- **Messages $\mathcal{M}$ (Inference):** The belief propagation mechanism described in Sec. 4.2, where node $j$ transmits prediction evidence to node $i$ weighted by edge strength.

- **Update Rules $\mathcal{U}$ (Evolution):** The temporal dynamics described in Sec. 4.3, which update the node states ($\mathbf{h}_i$) and consequently the edge topology ($\mathcal{E}^{(\ell+1)}$) for the next step.

This formalization highlights that our method is not merely a static graph regularization, but a closed-loop dynamic system.

## 4.2 CONFIDENCE-AWARE LOGIT REGULARIZATION

Building on the reliability-aware graph construction in Sec. 4.1, this module leverages class-wise confidence $R_j$ to inject reliability into the adjacency structure. In this sense, the logit regularization

is confidence-aware: predictions are not only smoothed by semantic neighbors but also weighted by their reliability.

To stabilize the discrete top-$k$ edge selection, we apply a sliding window average over the recent $L$ adjacency matrices. At test-time step $\ell$, the smoothed adjacency is

$$\bar{\mathbf{A}}^{(\ell)} = \frac{1}{L} \sum_{i=0}^{L-1} \mathbf{A}^{(\ell-i)}, \tag{8}$$

where $\mathbf{A}^{(\ell-i)}$ is the adjacency at step $\ell-i$. The buffer is updated online by enqueuing the latest matrix and discarding the oldest once its size exceeds $L$. This temporal smoothing avoids introducing an additional coefficient—only the window size $L$ is required—and makes the graph less sensitive to spurious pseudo-labels or abrupt prototype shifts.

Given a test input $\mathbf{x}$, we refine its raw prediction scores $\{p(y_i|\mathbf{x})\}_{i=1}^{C}$ by propagating them over the smoothed graph, we aggregate incoming messages to $y_i$ as:

$$p_{\text{graph}}(y_i|\mathbf{x}) = \frac{\sum_{y_j \in \mathcal{T}_i^k} \bar{\mathbf{A}}_{j,i}^{(\ell)} \, p(y_j|\mathbf{x})}{\sum_{y_j \in \mathcal{T}_i^k} \bar{\mathbf{A}}_{j,i}^{(\ell)}}, \tag{9}$$

where $\bar{\mathbf{A}}_{j,i}^{(\ell)}$ denotes the normalized weight of the edge from class $y_j$ to $y_i$ (row-normalized on $j$). Since the sliding-window averaging can slightly break exact row-normalization, the denominator in $p_{\text{graph}}$ re-normalizes the incoming weights. We then combine local and graph-based scores as

$$\hat{p}(y_i|\mathbf{x}) = \frac{p(y_i|\mathbf{x}) + p_{\text{graph}}(y_i|\mathbf{x})}{\sum_{m=1}^{C} \left( p(y_m|\mathbf{x}) + p_{\text{graph}}(y_m|\mathbf{x}) \right)}. \tag{10}$$

This process can be interpreted as one-step belief propagation in a class-level Markov random field, where local classifier outputs serve as node evidence and $\bar{\mathbf{A}}^{(\ell)}$ defines edge potentials. By coupling each class prediction with semantically and statistically reliable neighbors, the model reduces variance, suppresses noise, and mitigates semantic drift. The final pseudo-label is assigned as

$$y^*(\mathbf{x}) = \arg\max_{j} \hat{p}(y_j|\mathbf{x}). \tag{11}$$

### 4.3 GRAPH EVOLUTION

At each test-time step, SURE refines its predictions using the current PRG, while in turn updating the graph based on reliable pseudo-labels. This feedback loop enables mutual calibration: the graph constrains prediction dynamics via semantic uncertainty regularization, and reliable predictions reinforce graph consistency by refining class prototypes and local connectivity. These components form a dynamic regularization system that anchors adaptation to semantically trustworthy regions of the target domain.

**Prototype and reliability update.** Given a high-confidence pseudo-label $y^*(\mathbf{x})$ (*i.e.*, prediction confidence $> \theta$), we update the corresponding class prototype with a normalized moving average following Eq. 2. The update is tracked by a class-specific counter $N_i^{\text{proto}}$:

$$\mathbf{t}_i \leftarrow \text{Norm}\left( \frac{N_i^{\text{proto}} \cdot \mathbf{t}_i + \mathbf{f}}{N_i^{\text{proto}} + 1} \right), \quad N_i^{\text{proto}} \leftarrow N_i^{\text{proto}} + 1. \tag{12}$$

To assess the reliability of class $i$, we maintain a fixed-size sliding window buffer $\mathcal{Q}_i = \{c_i^{(1)}, \ldots, c_i^{(L)}\}$, which stores the most recent confidence scores $c(\mathbf{x})$ from test inputs assigned to class $i$. Only confidently assigned samples (confidence $\geq \theta$) contribute to the reliability buffer, using the same confidence as in $y^*(\mathbf{x})$. When a new score arrives, the oldest entry is removed if the buffer exceeds size $L$. The class-wise reliability statistics are then computed as:

$$\mu_i = \frac{1}{|\mathcal{Q}_i|} \sum_{n=1}^{|\mathcal{Q}_i|} c_i^{(n)}, \quad \sigma_i = \sqrt{\frac{1}{|\mathcal{Q}_i|} \sum_{n=1}^{|\mathcal{Q}_i|} \left( c_i^{(n)} - \mu_i \right)^2}. \tag{13}$$

---

**Algorithm 1:** Overview of the SURE algorithm.

---

**Input:** Pretrained CLIP, test stream $\{\mathbf{x}\}_{t=1}^{T}$, threshold $\theta$, neighbor size $k$, cache size $L$
**Output:** Adapted prediction $\hat{y}(\mathbf{x})$ for each input

1  Initialize reliability stats: $\mu_i \leftarrow 1.0$, $\sigma_i \leftarrow 0.0$, $\mathcal{Q}_i \leftarrow \emptyset$ for all $i$;
2  **for** *each test sample* $\mathbf{x}$ **do**
3      Compute CLIP scores $p(y_i|\mathbf{x})$;
4      Compute $\mathbf{S}_{ij} = \cos(\mathbf{t}_i, \mathbf{t}_j)$;
5      Compute $R_i = \mu_i \cdot (1 - \frac{\sigma_j}{\sigma_{\max}})$;
6      Form $\mathbf{W}_{i,j} = \mathbf{S}_{i,j} \cdot R_i \cdot R_j$;
7      Sparsify to top-$k$ neighbors $\Rightarrow \mathbf{A}^{(\ell)}$;
8      Maintain buffer $\{\mathbf{A}^{(\ell-i)}\}_{i=0}^{L-1}$;
9      Update: $\bar{\mathbf{A}}^{(\ell)} \leftarrow \frac{1}{L} \sum_{i=0}^{L-1} \mathbf{A}^{(\ell-i)}$;
10      Perform graph-based smoothing $\Rightarrow \hat{p}(y_i|\mathbf{x})$ (Eq. 10);
11      Assign pseudo-label $y^* \leftarrow \arg\max_i \hat{p}(y_i|\mathbf{x})$;
12      Compute confidence $c(\mathbf{x}) \leftarrow \hat{p}(y^*|\mathbf{x})$;
13      **if** $c(\mathbf{x}) > \theta$ **then**
14          Update prototype $\mathbf{t}_{y^*}$ via Eq. 12;
15          $N_{y^*}^{\text{proto}} \leftarrow N_{y^*}^{\text{proto}} + 1$;
16          Append $c(\mathbf{x})$ to $\mathcal{Q}_{y^*}$ and keep last $L$ entries;
17          Update $\mu_{y^*}, \sigma_{y^*}$ via Eq. 13;

---

Table 1: Results of natural distribution shifts for SURE and recent baselines using ResNet-50 and CLIP-ViT-B/16. We report the top-1 accuracy (%) for each dataset, along with the average accuracy for the five datasets and average OOD accuracy for ImageNet-A, -V2, -R, -Sketch. The best average and OOD average results are highlighted in **bold**.

| Method | ImageNet | ImageNet-A | ImageNet-V2 | ImageNet-R | ImageNet-Sketch | Average | OOD Average |
|---|---|---|---|---|---|---|---|
| CLIP-RN50 (Radford et al., 2021) | 59.81 | 23.24 | 52.91 | 60.72 | 35.48 | 46.43 | 43.09 |
| TPT (Shu et al., 2022) | 60.74 | 26.67 | 54.70 | 59.11 | 35.09 | 47.26 | 43.89 |
| TDA (Karmanov et al., 2024) | 61.35 | 30.29 | 55.54 | 62.58 | 38.12 | 49.58 | 46.63 |
| DPE (Zhang et al., 2024a) | 63.41 | 30.15 | 56.72 | 63.72 | 40.03 | 50.81 | 47.66 |
| BCA (Zhou et al., 2025) | 61.81 | 30.35 | 56.58 | 62.89 | 38.08 | 49.94 | 46.98 |
| R-TPT (Sheng et al., 2025) | 60.9 | 28.4 | 54.9 | 57.6 | 34.0 | 47.1 | 43.73 |
| **SURE (Ours)** | 64.08 | 29.57 | 57.75 | 63.38 | 40.83 | **51.12** | **47.88** |
| CLIP-ViT-B (Radford et al., 2021) | 68.34 | 49.89 | 61.88 | 77.65 | 48.24 | 61.20 | 59.40 |
| TPT (Shu et al., 2022) | 68.98 | 54.77 | 63.45 | 77.06 | 47.94 | 62.44 | 60.81 |
| MTA (Zanella & Ben Ayed, 2024) | 70.08 | 58.06 | 64.24 | 78.33 | 49.61 | 64.06 | 62.56 |
| TDA (Karmanov et al., 2024) | 69.51 | 60.11 | 64.67 | 80.24 | 50.54 | 65.01 | 63.89 |
| DPE (Zhang et al., 2024a) | 71.91 | 59.63 | 65.44 | 80.40 | 52.26 | 65.93 | 64.43 |
| ZERO (Farina et al., 2024) | 71.17 | 62.75 | 65.23 | 80.75 | 50.59 | 66.10 | 64.83 |
| TTL (Imam et al., 2025) | 70.23 | 60.51 | 64.55 | 77.54 | 48.61 | 64.29 | 62.80 |
| BCA (Zhou et al., 2025) | 70.22 | 61.14 | 64.90 | 80.72 | 50.87 | 65.37 | 64.16 |
| **SURE (Ours)** | 71.20 | 61.45 | 65.67 | 79.96 | 52.88 | **66.23** | **64.99** |

This buffered estimation provides a temporally smoothed and statistically stable summary of pseudo-label confidence for class $i$, avoiding fragile per-step updates. The resulting $(\mu_i, \sigma_i)$ are then used in Eq. 4 to update the reliability score $R_i$, thereby enabling robust graph evolution over time.

**Inference protocol.** Our overall process is shown in Algorithm 1. Final predictions are made using the adjusted posterior $\hat{p}(y_i|\mathbf{x})$, which integrates both the model's raw prediction and the structural consensus induced by the graph.

## 5 EXPERIMENT

### 5.1 EXPERIMENTAL SETUP

**Datasets.** We evaluate generalization under two complementary settings. For *natural distribution shifts*, we use ImageNet (Deng et al., 2009) and its OOD variants—ImageNet-V2 (Recht

Table 2: Results of cross-dataset generalization for SURE and recent baselines using ResNet-50 and CLIP-ViT-B/16. We report the top-1 accuracy (%) for each dataset, as well as the average accuracy across the ten datasets. The best average results are highlighted in **bold**.

| Method | SUN397 | Aircraft | EuroSAT | Cars | Food101 | Pets | Flower | Caltech | DTD | UCF101 | Average |
|---|---|---|---|---|---|---|---|---|---|---|---|
| CLIP-RN50 (Radford et al., 2021) | 60.85 | 16.11 | 25.79 | 55.89 | 74.82 | 82.97 | 62.77 | 87.26 | 40.37 | 59.48 | 56.63 |
| TPT (Shu et al., 2022) | 61.46 | 17.58 | 28.33 | 58.46 | 74.88 | 84.49 | 62.69 | 87.02 | 40.84 | 60.82 | 57.66 |
| TDA (Karmanov et al., 2024) | 62.53 | 17.61 | 42.11 | 57.78 | 77.75 | 86.18 | 68.74 | 89.70 | 43.74 | 64.18 | 61.03 |
| DPE (Zhang et al., 2024a) | 64.23 | 19.80 | 41.67 | 59.26 | 77.83 | 85.97 | 67.60 | 90.83 | 50.18 | 61.98 | 61.93 |
| BCA (Zhou et al., 2025) | 63.38 | 19.89 | 42.12 | 58.13 | 77.19 | 85.58 | 66.30 | 89.70 | 48.58 | 63.51 | 61.44 |
| **SURE (Ours)** | 64.18 | 20.45 | 41.41 | 62.07 | 79.78 | 85.35 | 68.56 | 89.98 | 50.90 | 65.24 | **62.79** |
| CLIP-ViT-B (Radford et al., 2021) | 65.63 | 23.22 | 50.42 | 66.11 | 82.86 | 86.92 | 66.99 | 93.55 | 45.04 | 65.16 | 64.59 |
| TPT (Shu et al., 2022) | 65.50 | 24.78 | 42.44 | 66.87 | 84.67 | 87.79 | 68.98 | 94.16 | 47.75 | 68.04 | 65.10 |
| MTA (Zanella & Ben Ayed, 2024) | 66.67 | 25.20 | 45.36 | 68.47 | 85.00 | 88.24 | 68.06 | 94.21 | 45.90 | 68.69 | 65.58 |
| TDA (Karmanov et al., 2024) | 67.62 | 23.91 | 58.00 | 67.28 | 86.14 | 88.63 | 71.42 | 94.24 | 47.40 | 70.66 | 67.53 |
| DPE (Zhang et al., 2024a) | 70.07 | 28.95 | 55.79 | 67.31 | 86.17 | 91.14 | 75.07 | 94.81 | 54.20 | 70.44 | 69.40 |
| Zero (Farina et al., 2024) | 66.90 | 24.42 | 43.77 | 68.48 | 84.58 | 87.20 | 66.82 | 94.14 | 45.86 | 68.57 | 65.07 |
| ZERO (Farina et al., 2024) | 67.63 | 25.21 | 42.17 | 68.97 | 86.77 | 87.83 | 67.17 | 94.41 | 45.86 | 69.18 | 65.52 |
| BCA (Zhou et al., 2025) | 68.41 | 28.59 | 56.63 | 66.86 | 85.97 | 90.43 | 73.12 | 94.69 | 53.49 | 67.59 | 68.59 |
| TTL (Imam et al., 2025) | 66.32 | 23.82 | 42.02 | 67.96 | 85.05 | 88.72 | 70.48 | 93.63 | 46.69 | 69.20 | 65.39 |
| **SURE (Ours)** | 70.82 | 28.92 | 53.60 | 69.31 | 87.47 | 89.81 | 77.75 | 94.89 | 55.26 | 72.56 | **70.04** |

Table 3: Comparison of our proposed SURE with baselines in terms of test time (s) and mean accuracy (%) on natural distribution shifts (ImageNet and its variants). Test time represents the average inference time per sample, measured on an NVIDIA RTX A6000 GPU.

| Method | Test Time (s) | Accuracy (%) | Δ Gain (%) |
|---|---|---|---|
| CLIP-ViT-B (Radford et al., 2021) | 0.004 | 61.20 | +0.00 |
| TPT (Shu et al., 2022) | 0.706 | 62.44 | +1.24 |
| MTA (Zanella & Ben Ayed, 2024) | 0.060 | 63.16 | +1.96 |
| DPE (Zhang et al., 2024a) | 0.189 | 65.93 | +4.73 |
| BCA (Zhou et al., 2025) | 0.023 | 65.37 | +4.17 |
| ZERO (Farina et al., 2024) | 0.082 | 66.10 | +4.90 |
| **SURE** | 0.067 | 66.23 | +7.12 |

et al., 2019), ImageNet-A (Hendrycks et al., 2021b), ImageNet-R (Hendrycks et al., 2021a), and ImageNet-Sketch (Wang et al., 2019). For *cross-dataset generalization*, we assess transfer to diverse domains, including objects (Caltech101 (Fei-Fei et al., 2004)), fine-grained categories (Oxford-Pets (Parkhi et al., 2012), StanfordCars (Krause et al., 2013), Flowers102 (Nilsback & Zisserman, 2008), Food101 (Bossard et al., 2014), FGVC-Aircraft (Du et al., 2020)), and scenes or textures (SUN397 (Xiao et al., 2010), EuroSAT (Helber et al., 2019), DTD (Cimpoi et al., 2014), UCF101 (Soomro et al., 2012)).

**Baselines.** We compare with representative TTA methods, including gradient-based approaches (TPT (Shu et al., 2022), TPS (Sui et al., 2024), DPE (Zhang et al., 2024a)) and gradient-free alternatives (BCA (Zhou et al., 2025), MTA (Zanella & Ben Ayed, 2024)). Among them, DPE, TDA, and BCA leverage historical test streams, while TPT and TPS perform online adaptation based solely on the current input.

**Implementation details.** We use ResNet-50 (RN50) (He et al., 2016) and ViT-B/16 (ViT-B) (Dosovitskiy et al., 2020) as CLIP image encoders (Radford et al., 2021), initialized with pretrained weights. All models are implemented using the official CLIP codebase. Following (Zhou et al., 2025), each class prototype is initialized with $N_i^{\text{proto}} = 30000$ confident samples and updated via normalized moving averages. For reliability estimation, we maintain a per-class sliding window of size $L = 5$. The graph neighbor size is set as $k = 3 \cdot \log(C)$, and the confidence threshold is $\theta = 0.3$. Prompts are derived by majority vote over 80 handcrafted templates (Radford et al., 2021).

## 5.2 COMPARISONS WITH STATE-OF-THE-ART

**Results on natural distribution shifts.** As shown in Tab. 1, **SURE** consistently outperforms prior methods on both RN50 and ViT-B backbones. The gains are most pronounced under RN50 (+4.79% over CLIP), where limited capacity amplifies semantic drift. Notably, SURE significantly improves

Table 4: Ablation study of key SURE components under natural distribution shifts on the ViT-B/16 backbone. Accuracy is reported for each dataset (%) along with average and OOD-only average across ImageNet-A/V2/R/Sketch. Each method is shown in two rows: the first for absolute accuracy, the second for the incremental gain compared to the preceding variant (or the baseline).

| Method | ImageNet | ImageNet-A | ImageNet-V2 | ImageNet-R | ImageNet-Sketch | Average | OOD Average |
|---|---|---|---|---|---|---|---|
| CLIP-ViT-B/16 | 68.34 | 49.89 | 61.88 | 77.65 | 48.24 | 61.20 | 59.40 |
| ProtoOnly | 69.31 | 57.92 | 63.40 | 78.48 | 50.13 | 63.85 | 62.48 |
| | 0.97 ↑ | 8.03 ↑ | 1.52 ↑ | 0.83 ↑ | 1.89 ↑ | 2.65 ↑ | 3.08 ↑ |
| +Graph w/o Rel | 69.63 | 57.68 | 63.80 | 78.63 | 50.70 | 64.09 | 62.70 |
| | 0.32 ↑ | -0.24 ↓ | 0.40 ↑ | 0.15 ↑ | 0.57 ↑ | 0.24 ↑ | 0.22 ↑ |
| +Graph + Rel | 70.07 | 60.49 | 64.36 | 78.17 | 52.73 | 65.16 | 63.94 |
| | 0.44 ↑ | 2.81 ↑ | 0.56 ↑ | -0.46 ↓ | 2.03 ↑ | 1.07 ↑ | 1.24 ↑ |
| +LogitProp (Full) | 71.20 | 61.45 | 65.67 | 79.96 | 52.88 | 66.23 | 64.99 |
| | 1.13 ↑ | 0.96 ↑ | 1.31 ↑ | 1.79 ↑ | 0.15 ↑ | 1.07 ↑ | 1.05 ↑ |

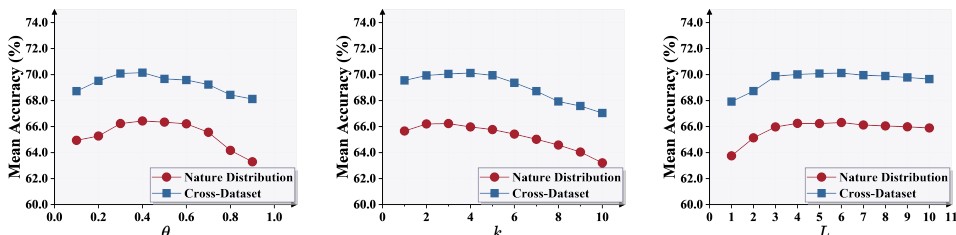

Figure 3: We show average accuracy under natural distribution shifts and cross-dataset generalization while varying confidence threshold $\theta$, neighbor size $k$, and buffer length $L$.

robustness on ImageNet-Sketch, a dataset characterized by abstract visual cues, reflecting its ability to stabilize adaptation when prototypes face severe drift. While the numerical margin over DPE on ImageNet-R is narrower—likely because low-level style cues are less influenced by semantic drift—SURE demonstrates superior stability across seeds. Under ViT-B, SURE reaches 66.23% average accuracy, exceeding strong baselines like ZERO and DPE, validating that structured semantic regularization provides complementary benefits beyond scaling backbone capacity.

**Results on cross-dataset generalization.** Tab. 2 shows that SURE achieves the highest average accuracy across ten diverse datasets (+5.49% over CLIP-ViT-B). Gains are especially notable in structure-sensitive domains (e.g., Aircraft, DTD) and fine-grained tasks (e.g., Flowers), indicating that reliability-aware propagation effectively suppresses misleading interactions in high-variance domains. In contrast, improvements on visually consistent datasets like Pets and Cars are relatively modest. This is likely because prototypes in these domains are already compact, leaving less room for graph-based refinement. Overall, SURE primarily benefits scenarios with ambiguous or highly variable semantics, where our structure-aware refinement offers the most value.

**Efficiency analysis.** Tab. 3 confirms that SURE offers a strong balance of performance and efficiency (66.23% accuracy at 0.067s/sample). It runs over 10× faster than TPT and outperforms ZERO (+0.13%) while reducing latency by ∼18%, avoiding the computational overhead of processing multiple augmented views. This efficiency stems from our lightweight design: graph updates scale linearly with class count $C$ and are amortized through a sliding buffer. Compared to BCA, SURE improves accuracy by +0.86% with negligible latency increase, confirming it is both deployment-friendly and robust in streaming adaptation.

## 5.3 ABLATION STUDY

We analyze key components of SURE using ViT-B/16 under natural distribution shifts.

**Effectiveness of model components.** As shown in Tab. 4, the minimal baseline (`ProtoOnly`) improves over raw CLIP by enabling continual adaptation. While adding graph structure (`+Graph w/o Rel`) incorporates semantic context, it can be risky if unreliable edges dominate; indeed, graph smoothing alone may even hurt performance in noisy domains. The reliability-aware modula-

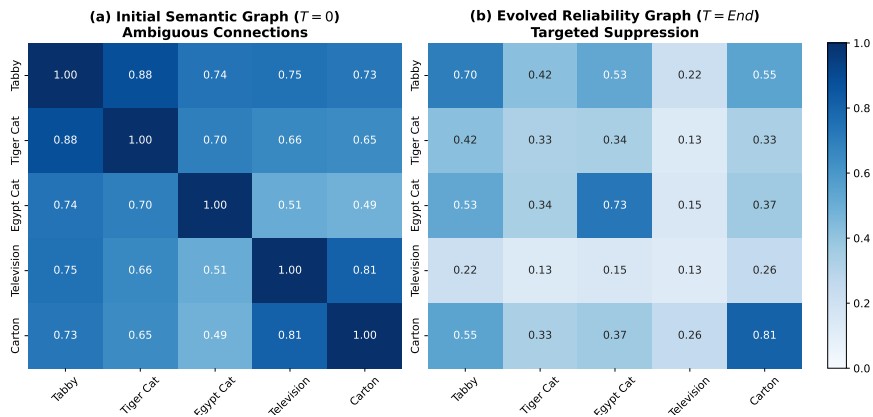

Figure 4: **Evolution of the Adjacency Matrix.** (Left) Initially, CLIP exhibits dense, ambiguous connections (e.g., 'Television' strongly connects to 'Tabby'). (Right) SURE suppresses unreliable outliers like 'Television' (weight drops to 0.13) while preserving valid semantic clusters.

tion (+`Graph + Rel`) solves this, boosting OOD accuracy by +1.24% via down-weighting noisy classes. Finally, enabling logit-level propagation (+`LogitProp`) yields the best result (+1.05%). The trajectory confirms that SURE's advantage comes from component synergy: the reliability mechanism corrects graph imbalances, and logit propagation consolidates information from statistically reliable neighbors to reduce prediction variance.

**Hyperparameter analysis.** Fig. 3 analyzes the confidence threshold $\theta$, neighbor size $k$, and window size $L$. Results show $\theta=0.4$ optimally balances noise suppression and information retention: lower thresholds admit unreliable classes, while higher ones discard too much signal. The optimal $k$ varies (4 for natural shifts, 3 for cross-dataset), reflecting semantic density: broader domains benefit from denser graphs, while fine-grained ones risk semantic leakage with large $k$. Performance stabilizes for $L>3$, indicating that a short buffer provides reliable estimation without excessive smoothing. These smooth variations confirm SURE's robustness to hyperparameter tuning.

### 5.4 VISUALIZATION OF GRAPH EVOLUTION

To validate our Prototype-Reliability Graph (PRG), we visualize adjacency matrix evolution on a "micro-universe" of 5 classes (three 'Cat' species, plus 'Television' and 'Carton') under simulated distribution shift.

**Results.** As shown in Fig. 4, the initial graph ($T = 0$) exposes CLIP's ambiguity, where the irrelevant 'Television' holds a dangerously strong connection to 'Tabby' (**0.75**), which could easily lead to error propagation during prototype updates. After adaptation ($T = End$), SURE's reliability gating identifies 'Television' as unstable due to high variance. Consequently, its influence is drastically attenuated, with its diagonal weight dropping to 0.13 and spurious connections suppressed. Crucially, the visualization reveals a clear hierarchy: the easy control class 'Carton' remains highly trusted (**0.81**), the hard fine-grained class 'Tiger Cat' is preserved but modulated, while the pure noise 'Television' is effectively silenced. This demonstrates that SURE learns a soft, statistically grounded topology rather than a simple hard threshold.

### 6 CONCLUSION

We presented SURE, a test-time adaptation framework that leverages a dynamic prototype-reliability graph to couple semantic similarity with class-wise reliability. By propagating structure-aware signals through this graph, SURE mitigates semantic drift and stabilizes predictions under distribution shifts. Extensive experiments confirm consistent gains across natural and cross-dataset benchmarks, highlighting both the effectiveness and efficiency of our closed-loop design. We believe SURE provides a principled foundation for semantic uncertainty modeling in TTA and can inspire future extensions toward graph-based reasoning in vision-language adaptation.

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

## A APPENDIX

This appendix provides additional details and analyses to complement the main paper. Specifically, we include:

- A detailed overview of the datasets used, including their label granularity and domain characteristics;

- The complete set of handcrafted textual prompts used for zero-shot and ensemble-based classification;

- An evaluation of prediction consistency across random seeds, highlighting the stability of SURE;

- A systematic hyperparameter study covering graph structure, confidence thresholds, and update window size;

- A comparative analysis of different prompt initialization strategies and their impact on adaptation;

- Calibration performance analysis using Expected Calibration Error (ECE) across all benchmark settings;

- Additional remarks summarizing the robustness and generality of the SURE framework across evaluation scenarios.

These supplementary results further support the effectiveness and robustness of the proposed SURE framework under various evaluation perspectives.

### A.1 DATASET OVERVIEW

To assess both robustness under natural distribution shifts and generalization to novel domains, we adopt a total of 15 standard benchmarks, consistent with prior TTA literature such as TPT (Shu et al., 2022). Tab. 5 outlines each dataset's label granularity and test size.

ImageNet Deng et al. (2009) serves as the canonical large-scale object classification dataset, with 1,000 diverse classes. ImageNet-V2 Recht et al. (2019) introduces subtle distribution shifts via resampling. ImageNet-A Hendrycks et al. (2021b) contains adversarially filtered, high-difficulty examples. ImageNet-R Hendrycks et al. (2021a) includes artistic renditions and stylized depictions of objects, while ImageNet-Sketch Wang et al. (2019) features hand-drawn black-and-white sketches.

Beyond the ImageNet family, Caltech101 Fei-Fei et al. (2004), StanfordCars Krause et al. (2013), Flowers102 Nilsback & Zisserman (2008), Food101 Bossard et al. (2014), and OxfordPets Parkhi et al. (2012) focus on fine-grained visual categorization within constrained domains. FGVCAircraft Du et al. (2020) emphasizes viewpoint and structural variation in aircraft types, while DTD Cimpoi et al. (2014) consists of texture-centric categories. SUN397 Xiao et al. (2010) presents a challenging scene recognition task with high intra-class variability. EuroSAT Helber et al. (2019) introduces modality shift through multispectral satellite imagery, and UCF101 Soomro et al. (2012) samples human action frames from video data, adding temporal and motion-related visual variability.

Together, these datasets offer a comprehensive evaluation protocol covering object-, texture-, scene-, and action-level recognition across diverse real-world domains.

### A.2 PROMPT TEMPLATE SPECIFICATION

Following the common practice in vision-language pretraining Radford et al. (2021), we adopt a fixed set of handcrafted textual prompts to encode class semantics. Specifically, our method's results are obtained under the 80-way prompt ensemble, where each prompt serves as a natural language template for class name insertion. The full list of these handcrafted templates, originally proposed in Radford et al. (2021), is reproduced in Tab. 6.

Table 5: Summary of datasets used in our evaluation, categorized by the nature of distribution shift. The first group reflects natural variants of ImageNet, while the second spans diverse domains to assess generalization.

| Type | Dataset | #Classes | #Test Samples |
|------|---------|----------|---------------|
| Natural Shift | ImageNet (Deng et al., 2009) | 1000 | 50,000 |
| | ImageNet-V2 (Recht et al., 2019) | 1000 | 10,000 |
| | ImageNet-A (Hendrycks et al., 2021b) | 200 | 7,500 |
| | ImageNet-R (Hendrycks et al., 2021a) | 200 | 30,000 |
| | ImageNet-Sketch (Wang et al., 2019) | 1000 | 50,889 |
| Cross-Domain | Caltech101 (Fei-Fei et al., 2004) | 100 | 2,465 |
| | OxfordPets (Parkhi et al., 2012) | 37 | 3,669 |
| | StanfordCars (Krause et al., 2013) | 196 | 8,041 |
| | Flowers102 (Nilsback & Zisserman, 2008) | 102 | 2,463 |
| | Food101 (Bossard et al., 2014) | 101 | 30,300 |
| | FGVCAircraft (Du et al., 2020) | 100 | 3,333 |
| | SUN397 (Xiao et al., 2010) | 397 | 19,850 |
| | DTD (Cimpoi et al., 2014) | 47 | 1,692 |
| | EuroSAT (Helber et al., 2019) | 10 | 8,100 |
| | UCF101 (Soomro et al., 2012) | 101 | 3,783 |

Table 6: Complete list of 80 handcrafted prompt templates used for generating text embeddings, following the standard CLIP ensemble Radford et al. (2021). The placeholder {} is substituted by class names during inference.

```
a bad photo of a {}              a photo of many {}                 a sculpture of a {}              a photo of the hard to see {}
a low resolution photo of the {} a rendering of a {}                graffiti of a {}                 a bad photo of the {}
a cropped photo of the {}        a tattoo of a {}                   the embroidered {}               a photo of a hard to see {}
a bright photo of a {}           a photo of a clean {}              a photo of a dirty {}            a dark photo of the {}
a drawing of a {}                a photo of my {}                   the plastic {}                   a photo of the cool {}
a close-up photo of a {}         a black and white photo of the {}  a painting of the {}             a painting of a {}
a pixelated photo of the {}      a sculpture of the {}              a bright photo of the {}         a cropped photo of a {}
a plastic {}                     a photo of the dirty {}            a jpeg corrupted photo of a {}   a blurry photo of the {}
a photo of the {}                a good photo of the {}             a rendering of the {}            a {} in a video game
a photo of one {}                a doodle of a {}                   a close-up photo of the {}       a photo of a {}
the origami {}                   the {} in a video game             a sketch of a {}                 a doodle of the {}
a origami {}                     a low resolution photo of a {}     the toy {}                       a rendition of the {}
a photo of the clean {}          a photo of a large {}              a rendition of a {}              a photo of a nice {}
a photo of a weird {}            a blurry photo of a {}             a cartoon {}                     art of a {}
a sketch of the {}               a embroidered {}                   a pixelated photo of a {}        itap of the {}
a jpeg corrupted photo of the {} a good photo of a {}               a plushie {}                     a photo of the nice {}
a photo of the small {}          a photo of the weird {}            the cartoon {}                   art of the {}
a drawing of the {}              a photo of the large {}            a black and white photo of a {}  the plushie {}
a dark photo of a {}             itap of a {}                       graffiti of the {}               a toy {}
itap of my {}                    a photo of a cool {}               a photo of a small {}            a tattoo of the {}
```

## A.3 STABILITY ACROSS RANDOM SEEDS

To assess the robustness of our method under varying initialization and test-time orderings, we report the standard deviation of performance across three independent runs in Tab. 7 and Tab. 8. SURE exhibits consistently low variance across both natural distribution shifts and cross-dataset generalization benchmarks. On the ImageNet variants, deviations remain below $0.3\%$, with particularly stable performance on ImageNet-Sketch and ImageNet-V2. Similarly, on the cross-domain benchmarks, the standard deviations for SURE-ViT-B do not exceed $0.3\%$, with most domains exhibiting fluctuations within $0.2\%$. These results confirm that SURE not only achieves strong accuracy but also maintains stable behavior across random seeds and domain shifts.

## A.4 HYPERPARAMETER TUNING

We perform a systematic grid search over the key hyperparameters in our framework. For the graph neighbor size $k$, we evaluate values from $\log C$ to $10 \log C$ in integer multiples. The confidence threshold $\theta$ is searched within the range $[0.1, 0.9]$ with a step size of $0.1$, while the temporal smoothing window $L$ is varied from 1 to 10 in steps of 1. All hyperparameters are selected based on performance on the ImageNet validation set and then fixed for evaluation on both the natural distribution shift benchmarks and the cross-dataset generalization tasks. Following Zhou et al. (2025), the initialization count $N_i^{\text{proto}}$ for each class prototype is set to a constant prior to adaptation.

Table 7: Performance of SURE on natural distribution shifts with standard deviation across three runs. Both ResNet-50 and ViT-B backbones are evaluated. SURE consistently improves accuracy over CLIP while maintaining low variance, indicating stable adaptation.

| Method | ImageNet | ImageNet-A | ImageNet-V2 | ImageNet-R | ImageNet-Sketch | Average | OOD Average |
|---|---|---|---|---|---|---|---|
| CLIP-RN50 (Radford et al., 2021) | 59.81 | 23.24 | 52.91 | 60.72 | 35.48 | 46.43 | 43.09 |
| **SURE (Ours)** | 64.08 ± .16 | 29.57 ± .18 | 57.75 ± .17 | 63.38 ± .28 | 40.83 ± .16 | **51.12** ± .12 | **47.88** ± .14 |
| CLIP-ViT-B (Radford et al., 2021) | 68.34 | 49.89 | 61.88 | 77.65 | 48.24 | 61.20 | 59.40 |
| **SURE (Ours)** | 71.20 ± .14 | 61.45 ± .23 | 65.67 ± .21 | 79.96 ± .13 | 52.88 ± .16 | **66.23** ± .11 | **64.99** ± .16 |

Table 8: Cross-dataset generalization results with standard deviation across three random seeds. SURE achieves state-of-the-art accuracy on all datasets while exhibiting small performance variation, demonstrating robustness across diverse domains and categories.

| Method | SUN397 | Aircraft | EuroSAT | Cars | Food101 | Pets | Flower | Caltech | DTD | UCF101 | Average |
|---|---|---|---|---|---|---|---|---|---|---|---|
| CLIP-RN50 (Radford et al., 2021) | 60.85 | 16.11 | 25.79 | 55.89 | 74.82 | 82.97 | 62.77 | 87.26 | 40.37 | 59.48 | 56.63 |
| **SURE (Ours)** | 64.18 ± .04 | 20.45 ± .15 | 41.41 ± .22 | 62.07 ± .21 | 79.78 ± .09 | 85.35 ± .15 | 68.56 ± .15 | 89.98 ± .16 | 50.90 ± .21 | 65.24 ± .14 | **62.79** ± .11 |
| CLIP-ViT-B (Radford et al., 2021) | 65.63 | 23.22 | 50.42 | 66.11 | 82.86 | 86.92 | 66.99 | 93.55 | 45.04 | 65.16 | 64.59 |
| **SURE (Ours)** | 70.84 ± .08 | 28.92 ± .17 | 53.60 ± .24 | 69.31 ± .27 | 87.47 ± .09 | 89.81 ± .14 | 77.75 ± .25 | 94.89 ± .24 | 55.26 ± .18 | 72.56 ± .17 | **70.04** ± .14 |

## A.5 PROMPT VARIATION ANALYSIS

To examine how different textual initializations influence test-time adaptation, we report the performance of our method under several representative prompt configurations. While our main results are based on 80-way prompt ensemble (`+Emsemble`), we additionally evaluate two alternatives: a standard handcrafted prompt (`+Basic`) and CoOp-learned prompt embeddings (`+CoOp`). This analysis provides insights into the sensitivity of adaptation performance with respect to prompt design and semantic expressiveness.

- `+Basic`: A single handcrafted prompt "`a photo of a [class]`", which is commonly used in CLIP-based zero-shot classification.
- `+CoOp`: A learned prompt embedding introduced in Zhou et al. (2022b), trained with 16-shot supervision on ImageNet and composed of four class-specific tokens. We directly use the pretrained prompt embeddings without additional tuning, following Zanella & Ben Ayed (2024).

As shown in Tab. 9, prompt initialization has a non-trivial impact on adaptation performance. Compared to the basic prompt, both `+CoOp` and `+Ensemble` consistently improve accuracy across all ImageNet variants. Notably, `+CoOp` achieves the highest accuracy on most individual datasets and leads in both overall and OOD averages, reflecting the advantage of learned prompt embeddings. Meanwhile, `+Ensemble` also performs strongly, benefiting from linguistic diversity and offering competitive robustness under domain shift. These results highlight that SURE is not limited to a specific prompt format and can flexibly adapt to various textual initializations. The ability to incorporate both handcrafted and learned prompts further demonstrates the generality and robustness of our framework in real-world settings.

## A.6 CALIBRATION ANALYSIS

In addition to accuracy, we assess model confidence calibration using the Expected Calibration Error (ECE) Naeini et al. (2015), a standard metric that quantifies the alignment between predicted probabilities and actual correctness. Formally, ECE is computed as:

$$\text{ECE} = \sum_{m=1}^{M} \frac{|B_m|}{N} \left| \text{acc}(B_m) - \text{conf}(B_m) \right|, \tag{14}$$

Table 9: Performance of SURE under different prompt initialization strategies on ImageNet and its OOD variants using the CLIP-ViT-B/16 backbone. `Basic` uses a single handcrafted prompt ("a photo of a [class]"), `CoOp` adopts learned prompts from Zhou et al. (2022b), and `Ensemble` averages 80 handcrafted prompts. `+Ensemble` yields the best average OOD accuracy, while `+CoOp` achieves the highest overall performance.

| Method | ImageNet | ImageNet-A | ImageNet-V2 | ImageNet-R | ImageNet-Sketch | Average | OOD Average |
|---|---|---|---|---|---|---|---|
| CLIP-ViT-B (Radford et al., 2021) | 68.34 | 49.89 | 61.88 | 77.65 | 48.24 | 61.20 | 59.40 |
| SURE (Basic) | 69.68 | 60.12 | 63.98 | 80.34 | 51.56 | 65.13 | 64.00 |
| SURE (CoOp) | 74.82 | 62.38 | 67.04 | 81.35 | 53.79 | 67.88 | 66.14 |
| SURE (Emsemble) | 71.20 | 61.45 | 65.67 | 79.96 | 52.88 | 66.23 | 64.99 |

Table 10: Comparison of classification accuracy ($\uparrow$) and Expected Calibration Error (ECE, $\downarrow$) on nature-shift distributions (ImageNet and its 4 variants) and cross-dataset generalization benchmarks. While CLIP shows strong calibration without adaptation, it lags in accuracy. Calibration-aware baselines such as SaLS Murugesan et al. (2024) and C-TPT Yoon et al. (2024) improve ECE moderately but sacrifice generalization. SURE achieves the best accuracy overall while maintaining competitive calibration.

| Method | ImageNet OOD | | Cross-dataset | |
|---|---|---|---|---|
| | Acc. $\uparrow$ | ECE $\downarrow$ | Acc. $\uparrow$ | ECE $\downarrow$ |
| CLIP-ViT-B(Radford et al., 2021) | 57.54 | 6.29 | 65.12 | 3.75 |
| TPT (Shu et al., 2022) | 60.93 | 10.8 | 60.08 | 9.90 |
| TPT (Shu et al., 2022) + C-TPT (Yoon et al., 2024) | 60.70 | 8.32 | 65.47 | 4.64 |
| TPT (Shu et al., 2022) + SaLS (Murugesan et al., 2024) | 60.89 | 10.86 | 64.59 | 9.88 |
| ProtoOnly (Tab. 4) | 63.85 | 11.23 | 67.83 | 9.86 |
| **SURE (Ours)** | 66.23 | 7.48 | 70.08 | 6.54 |

where $M$ denotes the number of confidence bins, $B_m$ is the $m$-th bin containing $|B_m|$ samples, $N$ is the total number of samples, $\text{acc}(B_m)$ is the empirical accuracy within the bin, and $\text{conf}(B_m)$ is the mean predicted confidence. Lower ECE values indicate better-calibrated predictions.

**Calibration-performance trade-off.** Tab. 10 compares SURE with several test-time adaptation methods in terms of both classification accuracy and model calibration, measured by ECE. The zero-shot CLIP baseline exhibits strong calibration due to its lack of adaptation dynamics, but its accuracy is notably limited, especially under domain shift. TPT Shu et al. (2022) improves accuracy via test-time prompt tuning, but significantly increases ECE. Notably, the ProtoOnly baseline achieves competitive accuracy gains but suffers from the highest calibration error (11.23 on ImageNet OOD), indicating that naive prototype updates without constraints tend to generate overconfident predictions and exacerbate semantic drift.

To mitigate miscalibration, calibration-aware extensions such as SaLS Murugesan et al. (2024) and C-TPT Yoon et al. (2024) introduce entropy-based or post-hoc smoothing mechanisms. While they reduce ECE to some extent, their performance on cross-dataset benchmarks remains inferior. In contrast, SURE achieves a superior balance: it consistently outperforms all baselines in accuracy across both OOD and cross-dataset settings, while keeping ECE relatively low (7.48 and 6.54, respectively). Crucially, SURE significantly reduces ECE compared to ProtoOnly (7.48 vs. 11.23), demonstrating that the proposed reliability-aware graph acts as a safety net, effectively suppressing noise to enhance predictive performance while preserving trustworthy confidence estimation.

## A.7 ADDITIONAL REMARKS

The extended results presented in this appendix offer further evidence for the effectiveness and robustness of the proposed SURE framework. Across all 15 datasets, SURE consistently improves accuracy under both natural distribution shifts and cross-domain generalization scenarios, while maintaining low variance and competitive calibration.

The stability analysis shows that our method performs reliably under different random seeds and test-time permutations, which is crucial for real-world deployment. The hyperparameter search confirms that SURE is not overly sensitive to tuning, especially when initialized using prior knowledge from zero-shot models.

Our prompt variation experiments indicate that SURE is not limited to a specific prompt initialization. While learned prompts such as CoOp can lead to strong performance, SURE also demonstrates robust adaptation when combined with handcrafted templates, particularly under distribution shifts. This flexibility highlights SURE's ability to accommodate diverse prompt configurations, benefiting from either semantic richness or data-driven optimization.

Moreover, the calibration results suggest that SURE mitigates the overconfidence typically caused by adaptation, preserving the trustworthiness of predictions. Taken together, these findings highlight the practicality of SURE as a simple, effective, and robust solution for adapting vision-language models without access to labeled target data.

We hope this supplementary analysis provides readers with deeper insights into the design choices and empirical behavior of our method.

