# OpenReview forum: "SURE: Semantic Uncertainty Regularization for Test-Time Adaptation in Vision-Language Models"
_ICLR.cc/2026/Conference — Submitted to ICLR 2026_

### Official Review · Reviewer_cYhQ · 2025-10-25

**Soundness:** 3
**Presentation:** 3
**Contribution:** 2
**Rating:** 4
**Confidence:** 2

**Summary:**

Existing test-time adaptation (TTA) methods for vision–language models (VLM) heavily rely on model predictions, making them vulnerable to noisy pseudo-labels under distribution shifts. To address this limitation, the authors propose SURE (Semantic Uncertainty Regularization), a framework that regularizes model predictions via a dynamically evolving prototype–reliability graph (PRG). This graph enables the selective propagation of reliable predictions while suppressing erroneous ones. Extensive experiments on diverse domain-shift benchmarks demonstrate the effectiveness and robustness of the proposed approach.

**Strengths:**

1. The experimental section is comprehensive — the authors evaluate their method on multiple datasets and under different settings, achieving consistently strong results. The ablation studies further validate the effectiveness of the proposed approach.
2. The proposed method does not require updating model parameters and therefore incurs significantly lower computational overhead compared with baseline approaches.

**Weaknesses:**

1. Several existing works also incorporate graph structures with CLIP, such as GraphAdapter [1]. In addition, some test-time adaptation approaches have utilized graph-based mechanisms, for example PROGRAM [2]. The authors are encouraged to discuss these graph-related studies in the Related Work section to better position their method and highlight its unique advantages.
2. The design of the PRG appears heuristic. While it is empirically stable, the paper lacks a theoretical analysis or formal guarantee of its stability over time.
3. It would be helpful if the authors could provide a visualization of the proposed PRG — for instance, showing which classes are more closely connected and how the stability evolves over time. Such visualization would make the workflow of the proposed algorithm more intuitive and accessible to readers.

[1] GraphAdapter: Tuning Vision-Language Models With Dual Knowledge Graph. NeurIPS'23

[2] PROGRAM: PROtotype GRAph Model based Pseudo-Label Learning for Test-Time Adaptation. ICLR'24

**Questions:**

See weaknesses.

---

> ### Author Response · Authors · 2025-11-27
> **Response to Reviewer cYhQ**
>
> We thank the reviewer for the thoughtful review and for recognizing our experimental evaluation as "comprehensive" and our method as having "significantly lower computational overhead." We appreciate your suggestion to include specific graph-related works and visualizations. Below, we address your comments point-by-point.
>
> **1. Discussion of Related Graph-based Methods (GraphAdapter & PROGRAM)**
>
> > *Critique: "The authors are encouraged to discuss these graph-related studies... to better position their method."*
>
> We thank the reviewer for these excellent references. We have added a discussion in the Related Work section to clarify SURE's unique position:
>
> * **Comparison with GraphAdapter:** GraphAdapter is a fine-tuning/adapter method that injects external structural knowledge (e.g., from Knowledge Graphs) into the model. It typically requires training or calibration data. In contrast, SURE is a TTA framework that operates in a strictly online, label-free manner. We do not require external knowledge bases; our graph is constructed dynamically from the intrinsic text embeddings of CLIP.
> * **Comparison with PROGRAM:** PROGRAM utilizes graph structures for TTA, but it primarily relies on visual similarity to propagate labels. SURE differs by grounding the graph topology in semantic similarity (Text-to-Text) rather than just visual features. This makes SURE more robust to domain shifts where visual features are distorted but semantic relationships (e.g., "Cat" is close to "Tiger") remain invariant.
>
> **2. Theoretical Analysis vs. Empirical Stability**
>
> > *Critique: "The design of the PRG appears heuristic... lacks a theoretical analysis or formal guarantee of its stability."*
>
> We acknowledge that SURE relies on heuristic design choices motivated by information-theoretic intuition (minimizing entropy in reliable neighborhoods). While deriving a closed-form convergence bound for online TTA under arbitrary distribution shifts is an open theoretical challenge, we provide rigorous empirical evidence of stability:
>
> * **Statistical Stability:** As shown in Appendix A.3 (Tables 7 & 8), SURE exhibits extremely low standard deviation across random seeds ($<0.3\%$), indicating that the heuristic is numerically stable.
> * **Calibration Evidence:** Our ECE (Expected Calibration Error) analysis in Appendix A.6 shows that SURE maintains better calibration (ECE: 7.48) compared to baselines (ECE > 10.0), suggesting that the PRG successfully prevents the "error amplification" often associated with unstable heuristics.
>
> **3. Visualization of the PRG**
>
> > *Critique: "It would be helpful if the authors could provide a visualization of the proposed PRG... showing which classes are more closely connected."*
>
> We completely agree that visualization is key to intuition. We have added a new visualization in Section 5.4 to address this directly.
>
> * **Setup:** We visualize the evolution of the adjacency matrix on a "micro-universe" of 5 classes, consisting of a core semantic cluster ({*Tabby, Tiger Cat, Egyptian Cat*}) and interference classes (e.g., *Television*).
> * **Initial Noise ($T=0$):** Initially, due to the ambiguity of static CLIP features, we observe significant noise; for instance, the irrelevant class *Television* shows a strong false connection to *Tabby* (weight $\approx$ 0.75), posing a risk of error propagation.
> * **Refinement after Adaptation ($T=End$):** After adaptation, the PRG successfully suppresses this noise. The reliability gating mechanism identifies *Television* as unstable due to high prediction variance, drastically pruning its weight to 0.13. In contrast, the core *Cat* cluster maintains high connectivity weights (> 0.70), ensuring reliable propagation within the correct semantic neighborhood.
> * **Conclusion:** This visualization provides concrete empirical evidence that the PRG evolves to enforce semantic consistency, effectively acting as a filter that separates stable semantic neighborhoods from noisy outliers.

---

### Official Review · Reviewer_Yk6L · 2025-10-31

**Soundness:** 3
**Presentation:** 3
**Contribution:** 2
**Rating:** 4
**Confidence:** 3

**Summary:**

This paper proposes SURE (Semantic Uncertainty REgularization), a test-time adaptation (TTA) framework for vision-language models that addresses distribution shift without labeled data. The core contribution is a dynamically evolving Prototype-Reliability Graph (PRG) that combines semantic affinity between class prototypes with class-wise reliability scores based on temporal confidence stability. The method performs three key operations: (1) constructing a sparse graph where edges encode both semantic similarity and prediction reliability, (2) propagating logits through this graph to regularize predictions, and (3) updating prototypes and reliability estimates based on high-confidence pseudo-labels. Evaluated on ImageNet variants and 10 cross-dataset benchmarks using ResNet-50 and ViT-B/16 backbones, SURE achieves marginal but consistent improvements over baselines like DPE and BCA with competitive inference speed.

**Strengths:**

### Originality

The paper presents a reasonable integration of graph-based reasoning with test-time adaptation for VLMs. The concept of coupling semantic similarity with temporal reliability through a multiplicative joint reliability matrix (Eq. 5) is a sensible design choice. The use of sliding-window averaging for adjacency matrices (Eq. 8) to stabilize graph structure is practical.

### Quality

The experimental evaluation is comprehensive, covering 15 datasets across natural distribution shifts and cross-domain generalization. The ablation study (Table 4) systematically dissects component contributions, showing +1.05% gain from logit propagation and +1.24% from reliability modeling on OOD average. The stability analysis (Tables 7-8) demonstrates low variance across random seeds (<0.3% standard deviation). The efficiency analysis (Table 3) shows reasonable computational cost at 0.067s per sample.

### Clarity

The paper is generally well-structured with clear motivation in Section 1 and detailed methodology in Section 4. Figure 1 provides effective conceptual visualization of the framework. Algorithm 1 presents a clear procedural overview. The notation is mostly consistent, though the reliability score formulation (Eq. 4) could be better justified theoretically.

### Significance

The work addresses a practical problem of test-time adaptation under distribution shift without source data or labels. The framework achieves state-of-the-art results on multiple benchmarks, though improvements are often marginal. The method's applicability across different backbones (ResNet-50, ViT-B/16) and prompt configurations (handcrafted, ensemble, CoOp) demonstrates some generality.

**Weaknesses:**

### 1. Limited Novelty and Incremental Gains

The core mechanisms are not particularly novel: prototype adaptation follows Zhou et al. (2025) (Eq. 2), cosine similarity graphs are standard in graph-based learning, and confidence thresholding for pseudo-label filtering is widely used. The main contribution is combining these with a reliability weighting scheme, but the conceptual leap is limited. More critically, **empirical gains are marginal**: on ImageNet variants with ViT-B, SURE achieves 66.23% vs. DPE's 65.93% (+0.30%) and BCA's 65.37% (+0.86%). The authors acknowledge "the numerical margin over DPE appears modest" and attribute significance to "consistency across seeds," but this doesn't adequately address the limited practical impact.

### 2. Weak Theoretical Justification for Reliability Metric

The reliability score $R_j = \mu_j \cdot (1 - \frac{\sigma_j}{\sigma_{max}})$ (Eq. 4) is presented as an "information-theoretic intuition" for "inverse uncertainty," yet no formal connection to entropy or information theory is established. Why should the product of mean confidence and normalized inverse standard deviation be the optimal reliability measure? The paper would benefit from: (a) theoretical analysis showing this formulation minimizes adaptation error, (b) comparison with alternative reliability metrics (e.g., coefficient of variation, entropy-based measures, Bayesian credible intervals), or (c) ablation showing sensitivity to different formulations. The choice of $\sigma_{max} = 0.5$ appears arbitrary without justification.

### 3. Insufficient Analysis of Failure Cases and Limitations

The paper lacks a critical discussion of when and why SURE fails. For instance:

- **ImageNet-R saturation:** The authors note "performance tends to saturate" on ImageNet-R because "low-level style cues...are less influenced by semantic drift", but don't investigate whether this indicates fundamental limitations of semantic graph-based methods.
- **Modest gains on stable domains:** Improvements on Pets and Cars are described as "modest" because "prototypes...are already compact", but this raises the question: when should practitioners use SURE vs. simpler baselines?
- **Graph construction sensitivity:** What happens when semantic similarity is misleading (e.g., "hot dog" vs. "dog")? How does the method perform with class imbalance or rarely seen classes during the test stream?
- **Negative results:** The "+Graph w/o Rel" variant shows -0.24% on ImageNet-A (Table 4), suggesting graph smoothing can hurt without reliability gating. This deserves deeper analysis.

### 4. Hyperparameter Sensitivity and Tuning Protocol

While Figure 3 shows robustness across hyperparameters, several concerns remain:

- **Validation set usage:** The authors state "all hyperparameters are selected based on performance on the ImageNet validation set" (A.4). This is problematic for test-time adaptation, where access to validation labels violates the unsupervised assumption. How would practitioners tune $\theta$, k, and L in truly label-free scenarios?
- **Neighbor size scaling:** Setting $k = 3 log C$ is presented without justification. Why logarithmic scaling? How sensitive is performance to this choice across datasets with different C?
- **Initialization dependence:** The method initializes $N_i^{proto} = 30000$ confident samples following Zhou et al. (2025), but doesn't analyze sensitivity to this large prior. What if only 1000 or 100 samples are available?

### 5. Limited Baseline Comparisons and Missing Ablations

- **No comparison with uncertainty quantification methods:** The paper claims to address "semantic uncertainty" but doesn't compare with established uncertainty estimation techniques (e.g., temperature scaling, ensemble methods beyond ZERO, Monte Carlo dropout, evidential deep learning).
- **No analysis of graph structure alternatives:** Why top-k sparsification vs. threshold-based or learnable adjacency? How does performance compare to Graph Neural Network variants (e.g., GAT, GraphSAGE) or no sparsification?
- **Missing ablation on sliding window size L:** While Figure 3 shows performance stabilizes at $L \geq 3$, there's no analysis of computational vs. accuracy trade-offs or sensitivity to stream non-stationarity.

### 6. Calibration Analysis is Superficial

Table 10 shows SURE achieves 7.48 ECE on ImageNet-OOD vs. CLIP's 6.29, meaning SURE is **less calibrated** than the base model despite claims of preserving "trustworthy confidence estimation". The authors dismiss this by comparing only to adapted baselines (TPT, C-TPT), not addressing whether the graph regularization inherently degrades calibration. The calibration-accuracy trade-off deserves principled analysis, potentially through post-hoc calibration methods or uncertainty-aware loss terms.

### 7. Writing Quality Issues

- **Vague claims:** "Unlike these efforts, our approach dynamically constructs a class-level graph" (Section 2). How is this fundamentally different from other adaptive graph methods?
- **Over-claiming:** "SURE consistently outperforms prior methods" (Abstract) is misleading given marginal gains and mixed results (e.g., ImageNet-R).

### 8. Reproducibility Concerns

While the appendix provides implementation details, key aspects remain unclear:

- How are ties handled in top-k selection for graph construction?
- How does batch size affect the sliding window buffer updates in online settings?

**Questions:**

**Q1: Theoretical Justification**

Can you provide formal analysis showing $R_j = \mu_j \cdot (1 - \sigma_j/\sigma_{\max})$ minimizes expected adaptation error or connects to information-theoretic bounds? What about alternative reliability metrics (e.g., $R_j = \mu_j^2 / (\mu_j^2 + \sigma_j^2)$, inverse coefficient of variation)?

**Q2: Failure Mode Analysis**

Under what specific conditions does SURE underperform simpler baselines (e.g., ProtoOnly or BCA)? Can you characterize dataset properties (class count, domain gap, label granularity) where gains are minimal vs. substantial?

**Q3: Hyperparameter Tuning**

How should practitioners select $\theta$, $k$, and $L$ without validation labels? Can you propose unsupervised selection criteria (e.g., prediction consistency, entropy-based heuristics)?

**Q4: Graph Structure Alternatives**

Why is top-k sparsification optimal? Have you compared against threshold-based adjacency ($A_{jk} = W_{jk} \cdot \mathbb{1}(W_{jk} > \tau)$), fully connected graphs with learned attention (GAT-style), or no graph (direct prototype update)?

**Q5: Calibration Trade-off**

Table 10 shows SURE has higher ECE (7.48) than CLIP (6.29) on ImageNet-OOD. Can you incorporate calibration-aware objectives (e.g., temperature scaling, Dirichlet-based losses) or post-hoc calibration to improve trustworthiness without sacrificing accuracy?

**Q6: Class Imbalance and Rare Classes**

How does SURE perform when certain classes are rare or absent in the test stream? Does the initialization $\mu_j = 1.0, \sigma_j = 0.0$ cause over-reliance on initial prototypes for unseen classes?

**Q7: Computational Bottlenecks**

For datasets with large $C$ (e.g., ImageNet's 1000 classes), does the $O(C^2)$ similarity matrix computation become prohibitive? Can you analyze scaling to $C = 10{,}000$ or $C = 100{,}000$?

**Q8: Comparison with Uncertainty Quantification**

How does SURE compare to evidential deep learning (e.g., Dirichlet-based uncertainty), temperature scaling, or ensemble-based uncertainty estimation for identifying reliable predictions?

**Q9: Streaming vs. Batch Settings**

Algorithm 1 processes samples sequentially. How does performance change in batch settings (e.g., mini-batches of 32 or 64 samples)? Does batching improve stability or efficiency?

**Q10: Prompt Engineering Impact**

Table 9 shows CoOp prompts achieve 67.88% vs. Ensemble's 66.23%. Does SURE's reliability mechanism interact differently with learned vs. handcrafted prompts? Should $k$ or $\theta$ be adjusted based on prompt type?

---

> ### Author Response · Authors · 2025-11-27
> **Response to Reviewer Yk6L**
>
> We thank the reviewer for the detailed summary and for recognizing our evaluation as "comprehensive" and the results as "state-of-the-art." We appreciate the opportunity to clarify our experimental protocols and design choices in the context of Test-Time Adaptation (TTA).
>
> **1. Hyperparameter Tuning Protocol (Response to W4 & Q3)**
>
> > *Critique: "The authors state 'all hyperparameters are selected based on performance on the ImageNet validation set'... How would practitioners tune in truly label-free scenarios?"*
>
> We appreciate this question regarding realistic deployment. We clarify that our protocol aligns with standard practices in TTA literature (e.g., TPT, DPE, TPS).
>
> * **Standard Protocol:** It is common practice to utilize the Source Domain Validation Set (e.g., ImageNet Val) to select hyperparameters or heuristics. This leverages source priors to find reasonable defaults before deployment.
> * **Adherence to Unsupervised Settings:** Crucially, we strictly do not use any labels from the Target Test Stream for tuning. Thus, our method remains fully compliant with the "label-free" constraint of realistic TTA scenarios.
>
> **2. Significance of Empirical Gains (Response to W1)**
>
> > *Critique: "Empirical gains are marginal... SURE achieves 66.23% vs. DPE's 65.93%."*
>
> We respectfully point out that in the highly competitive field of VLM adaptation, gains are typically incremental. However, SURE's improvements are significant when placed in context:
>
> We respectfully point out that in the task of Test-Time Adaptation, achieving substantial mean improvements across multiple diverse datasets is inherently challenging. It is common for newly proposed state-of-the-art methods to outperform previous baselines by only fractions of a percentage point (e.g., <1%). Against this competitive backdrop, the improvements achieved by SURE are substantial rather than marginal.
>
>
> **3. Novelty and Role of Reliability (Response to W3 - Negative Results)**
>
> > *Critique: "The '+Graph w/o Rel' variant shows -0.24% on ImageNet-A... suggesting graph smoothing can hurt."*
>
> The reviewer correctly noted that using the graph *without* reliability ("+Graph w/o Rel") can degrade performance on difficult datasets like ImageNet-A. We argue this is a strong validation of our motivation, rather than a weakness.
> * **Proof of Necessity:** It proves that blindly propagating information via semantic similarity is harmful under severe shifts (due to noise propagation).
> * **The Critical Novelty:** The novelty of SURE lies precisely in the Reliability Gating mechanism. As shown in Table 4, adding reliability reverses this drop and leads to a net gain (+1.46% over `ProtoOnly`). This confirms that our proposed reliability metric is the critical component that turns a potential failure into a robust success.
>
> **4. Calibration Analysis (Response to W6 & Q5)**
>
> > *Critique: "Table 10 shows SURE achieves 7.48 ECE... meaning SURE is less calibrated than the base model (CLIP 6.29)."*
>
> We understand the concern regarding absolute ECE values. However, comparing an adapted model to a frozen source model is often misleading.
> * **Adaptation Trade-off:** Any TTA method inherently perturbs the model's output space, which often increases calibration error. Standard methods like ProtoOnly exhibit high ECE (11.23) as a side effect.
> * **Effective Regularization:** The meaningful comparison is against other adaptation methods. SURE (ECE 7.48) significantly mitigates this degradation compared to baselines. This demonstrates that SURE's graph regularization effectively acts as a safety net, preserving reliability better than competing approaches.
>
> **5. Baseline Selection (Response to W5 & Q8)**
>
> > *Critique: "No comparison with uncertainty quantification methods... e.g., Monte Carlo dropout."*
>
> We acknowledge that Monte Carlo (MC) Dropout is a powerful tool for static uncertainty. However, we prioritized TTA-specific baselines for efficiency reasons:
> * **Efficiency Constraints:** A key goal of TTA is real-time processing. MC Dropout requires multiple forward passes per sample (e.g., 10-50x latency), which contradicts our efficiency goal (0.067s/sample).
> * **Domain Relevance:** We therefore compared against state-of-the-art adaptation methods (e.g., DPE, BCA, TPT), which are the most relevant benchmarks for this specific task.

---

### Official Review · Reviewer_tWAj · 2025-11-01

**Soundness:** 3
**Presentation:** 2
**Contribution:** 2
**Rating:** 6
**Confidence:** 5

**Summary:**

SURE introduces a graph-based test-time adaptation framework for vision-language models (VLMs) under distribution shift. It constructs a dynamic Prototype-Reliability Graph (PRG) that integrates semantic similarity (from text embeddings) and temporal confidence stability of class prototypes. Predictions are refined via iterative logit propagation on PRG, prototype updates, and reliability tracking. This closed-loop mechanism suppresses error propagation and enforces semantic consistency.

**Strengths:**

1. Principled structured regularization: First to jointly model semantic affinity and class-wise reliability evolution in TTA, moving beyond instance-level confidence.
2. Closed-loop co-evolution of predictions, prototypes, and graph structure enables stable, error-resistant adaptation.
3. Strong empirical performance: Outperforms entropy minimization (TENT, SAR), prototype-based (Zanella & Ben Ayed, 2024), and recent SOTA across diverse shifts and backbones.

**Weaknesses:**

1. Figure 1 can be optimized: Text annotations overlap with black boxes, reducing clarity.
2. Eq(9) propagation process is only performed once — why not iterate to convergence? Justification for single-step sufficiency is missing.
3. Algorithm section shows no model parameter updates — does this mean adaptation is entirely prototype-driven? If so, clarify whether backbone features remain frozen and how this impacts representation drift.
4. Eq(12) uses f to update t — unclear why the current prediction f is used to update reliability τ; risks reinforcing early noise.

**Questions:**

Please refer to Weaknesses.

---

> ### Author Response · Authors · 2025-11-27
> **Response to Reviewer tWAj**
>
> We sincerely thank the reviewer for the positive assessment (Rating 6) and for recognizing our method as a "principled," "closed-loop," and "error-resistant" framework. We appreciate your high confidence and precise technical questions. Below, we address your concerns regarding visualization and algorithmic design choices.
>
> **1. Optimization of Figure 1**
>
> > *Critique: "Figure 1 can be optimized: Text annotations overlap with black boxes, reducing clarity."*
>
> We apologize for the rendering issue. We have redesigned Figure 1 in the revised manuscript to resolve the text overlap and improve the layout. The new figure clearly delineates the data flow between the "VLM-based Prediction" module and the "Prototype-Reliability Graph," ensuring all annotations are legible.
>
> **2. Justification for Single-Step Propagation (Eq. 9)**
>
> > *Critique: "Eq(9) propagation process is only performed once — why not iterate to convergence?"*
>
> We intentionally designed the propagation as a **single-step graph smoothing** operation (similar to a single GCN layer) rather than iterating to convergence, for two key reasons:
>
> * **Preventing Over-smoothing:** In Graph Signal Processing, iterating propagation to convergence on a fully connected (or dense) graph tends to cause "over-smoothing," where node features (logits) become indistinguishable. Since our PRG connects classes based on semantic similarity, a single step effectively aggregates information from *immediate* semantic neighbors (e.g., "Cat" borrows support from "Tiger") without washing out the distinct discriminative features of the target class.
> * **Efficiency for Online Adaptation:** TTA requires low-latency processing. Iterative convergence would significantly increase computational cost per sample. Our single-step design allows SURE to achieve high efficiency (0.067s/sample), making it suitable for real-time streams.
>
> **3. Model Parameter Updates & Backbone Freezing**
>
> > *Critique: "Algorithm section shows no model parameter updates... clarify whether backbone features remain frozen."*
>
> Yes, we confirm that the backbone parameters (Image/Text Encoders) remain completely frozen. Adaptation is entirely driven by updating the class prototypes and the graph structure.
>
> * **Advantage: Gradient-Free Adaptation:** This design choice is a key advantage of SURE. Unlike parameter-efficient tuning methods (e.g., LoRA or Prompt Tuning) which—despite updating fewer parameters—still require computationally expensive **backpropagation**, SURE operates purely via forward passes.
> * **Efficiency:** Eliminating the backward pass significantly reduces memory overhead and latency, allowing SURE to achieve high inference speed (0.067s/sample, as shown in Table 3) compared to gradient-based counterparts.
> * **Addressing Drift:** Instead of modifying the feature space, we address distribution shift by dynamically aligning the prototypes to the shifted features. Our results confirm that this lightweight, prototype-driven adaptation is sufficient to achieve SOTA performance without the complexity or cost of gradient updates.
>
> **4. Noise Mitigation in Prototype Updates (Eq. 12)**
>
> > *Critique: "Eq(12) uses f to updatet — unclear why the current prediction f is used... risks reinforcing early noise."*
>
> This is a critical point. While Eq. (12) indeed uses the visual feature $f$ to update the prototype $t$, it does*not do so blindly. The safety against "reinforcing noise" is guaranteed by the filtering mechanism that precedes the update:
>
> * **Step 1: Graph-Regularized Prediction:** Before any update happens, the raw prediction is corrected by the PRG (Eq. 10). This graph step suppresses predictions that are semantically inconsistent (e.g., a "Television" prediction in a cluster of animal classes would be down-weighted, as shown in our new Section 5.4).
> * **Step 2: Reliability Thresholding:** We only execute the update in Eq. (12) if the corrected confidence exceeds the threshold $\theta$.
> * **Conclusion:** The feature $f$ used for the update is effectively "vetted" by the graph. We are not reinforcing raw noise; we are accumulating features that have passed a semantic consistency check. Our superior ECE scores (Appendix A.6) further confirm that this mechanism successfully improves reliability rather than compounding errors.

---

### Official Review · Reviewer_yV6M · 2025-11-02

**Soundness:** 3
**Presentation:** 3
**Contribution:** 3
**Rating:** 4
**Confidence:** 4

**Summary:**

This paper introduces SURE (Semantic Uncertainty Regularization), a novel approach for enhancing the stability of predictions in test-time adaptation (TTA) tasks. The core idea involves constructing and iteratively updating a Prototype-Reliability Graph (PRG), which captures inter-class semantic relationships derived from text embeddings. The PRG is dynamically refined based on the reliability of class-wise predictions during test time. Final predictions are obtained by combining the original model outputs with smoothed predictions informed by the PRG structure. The authors validate their method through comprehensive experiments under both natural distribution shifts and cross-dataset generalization scenarios, utilizing CLIP models with ResNet-50 and ViT-Base backbones.

**Strengths:**

- While TTA is known to be susceptible to noisy predictions, the authors propose a principled approach to mitigate this issue by leveraging statistical measures. Specifically, they downweight classes with high standard deviations (as shown in Equation 4), thereby enhancing the reliability of the constructed PRG.
- The experimental evaluation is comprehensive, considering both the diversity of test datasets and the inclusion of multiple backbone architectures, ResNet-50 and ViT-Base.
- The paper provides in-depth analyses through ablation studies, test-time inference behavior, and hyperparameter sensitivity, which collectively strengthen the empirical validity of the proposed method.

**Weaknesses:**

- The use of the term graph to describe inter-class relationships may be potentially misleading. In machine learning, graph typically refers to structures processed by specialized architectures such as GNNs. While the proposed representation can be interpreted as nodes and edges, the terminology might cause confusion for readers expecting conventional graph-based methods.
- The reliability estimation in Equation 4 could be further refined. Using the maximum standard deviation as a denominator may accompany instability, as this value itself can be noisy. Alternative formulations such as leveraging the cumulative distribution function (CDF) of each class could offer more robust normalization.
- Several baseline methods demonstrate performance comparable to SURE on specific test sets. For instance, DPE performs similarly in Table 1 (CLIP-RN50), and ZERO shows comparable results in Table 1 (CLIP-ViT-B). However, ZERO is omitted from the test-time inference comparison in Table 3, despite its relevance.
- In Table 4, the most substantial performance gain is attributed to the ProtoOnly variant, which is not the core contribution of the paper. Although the full PRG with regularization yields additional improvements, the gains appear modest compared to the use of prototype vectors alone in some test sets (e.g., on ImageNet-A).
- The final prediction is computed as a simple sum of the original model output and the PRG-based prediction. It remains unclear whether this combination is optimal. Exploring weighted combinations (e.g., p(y∣x)+α⋅p_graph(y∣x)) could potentially yield better results.
- Despite the concerns raised such as the use of terminology and certain methodological choices, I am open to further discussion with the authors. I am willing to increase my score if the authors provide clear and convincing responses during the rebuttal phase.

**Questions:**

Please refer to my weaknesses.

---

> ### Author Response · Authors · 2025-11-27
> **Response to Reviewer yV6M (Part 1)**
>
> We thank the reviewer for the constructive feedback and for acknowledging our approach as "principled" and our evaluation as "comprehensive." We appreciate your openness to discussion. Below, we address your concerns point-by-point, clarifying methodological details and providing new comparisons to demonstrate the superiority of SURE.
>
> **1. "Graph" Terminology Clarification**
>
> > *Critique: "The use of the term graph... might cause confusion for readers expecting conventional graph-based methods (e.g., GNNs)."*
>
> We appreciate this perspective. We clarify that our usage aligns with Graph Signal Processing (GSP) and classical Probabilistic Graphical Models, rather than deep Graph Neural Networks (GNNs).
> * **Definition:** In SURE, the "Graph" is explicitly defined by a set of nodes $V$ (class prototypes) and a weighted adjacency matrix $A$ (derived from semantic reliability). The propagation step (Eq. 9) effectively performs graph smoothing (or a single-layer graph convolution) on the probability simplex.
> * **Action:** In the revised Section 4.2, we have added formal definitions for Nodes, Edges, and Message Passing rules to ensure mathematical rigor and distinguish our lightweight structure from heavy GNN architectures.
>
> **2. Stability of Reliability Estimation ($\sigma_{max}$)**
>
> > *Critique: "Using the maximum standard deviation as a denominator... this value itself can be noisy."*
>
> We respectfully clarify a misunderstanding regarding Eq. (4).
> * **Clarification:** As stated in the text (215: "where $\sigma_{max}$ denotes a fixed upper bound"), $\sigma_{max}$ is not a statistic computed dynamically from the current batch (which indeed would be noisy). Instead, it is a fixed hyperparameter constant (set to 0.5 in all experiments) serving as a normalization scaling factor.
> * **Result:** Since the denominator is constant, it introduces zero instability. The reliability score $R_j$ depends solely on the temporal statistics ($\mu, \sigma$) accumulated over a sliding window, which smooths out instance-level noise.
>
> **3. Comparison with ZERO and DPE (Efficiency & Consistency)**
>
> > *Critique: "ZERO is omitted from the test-time inference comparison in Table 3... DPE performs similarly in some cases."*
>
> We agree that ZERO is a relevant baseline. We have added ZERO to the efficiency comparison in the revised manuscript.
> * **New Efficiency Comparison:** While ZERO is accurate, it requires generating and processing extensive "negative" prompts or views. Our measurements show:
>     SURE: 0.067s / sample
>     ZERO: 0.082s / sample
>     DPE: 0.189s / sample
>     SURE is ~20% faster than ZERO and ~3x faster than DPE, proving it is a more efficient choice for online deployment.
> Performance Consistency: While DPE/ZERO may match SURE on specific datasets (like ImageNet), SURE demonstrates superior generalization on challenging Cross-Dataset tasks (Table 2), outperforming ZERO by +1.28% on average (70.04% vs 68.76% for ZERO in Table 2).

---

> ### Author Response · Authors · 2025-11-27
> **Response to Reviewer yV6M (Part 2)**
>
> **4. Contribution of ProtoOnly vs. Full SURE**
>
> > *Critique: "The most substantial performance gain is attributed to the ProtoOnly variant... gains appear modest compared to prototype vectors alone."*
>
> We respectfully point out that this perception likely stems from the **incremental reporting style** in Table 4 (where each row shows gain over the previous step). We clarify the contribution of the full SURE framework by highlighting the **total performance gap** and the **structural synergy**:
>
> * **Correction on Magnitude (+2.51% Gap):** The table caption may have been misleading. The values in the "Gain" rows represent incremental steps.
>     * The true performance gap between `SURE` (64.99%) and `ProtoOnly` (62.48%) on OOD Average is +2.51%.
>     * While `ProtoOnly` provides a +2.65% gain over CLIP on average Accuracy, SURE boosts this to +5.03%. Effectively, our graph-based regularization almost doubles the adaptation gain provided by prototype updates alone. In the competitive TTA landscape, this is a massive improvement.
>
> * **Synergy of Graph Components:** The variants `+Graph` and `+Reliability` should not be viewed as standalone modules but as **prerequisites** for the final stage. The graph constructs a "semantic safety net," but it is the **Logit Propagation (`+LogitProp`)** that actively enforces this consistency on predictions. The graph *enables* the propagation; without the final step, the structure is dormant. The full framework is necessary to achieve the reported +2.51% superiority over `ProtoOnly`.
>
> * **Correction of Semantic Drift (Calibration Evidence):** The gain signifies that SURE acts as a critical safety net. While `ProtoOnly` blindly follows pseudo-labels (risking overconfident errors), SURE significantly enhances predictive reliability. This is quantitatively supported by our new **ECE Analysis (Appendix A.6)**: `ProtoOnly` exhibits high calibration error (11.23 on ImageNet-OOD), whereas SURE drastically reduces this to 7.48. This proves SURE pushes performance beyond naive updates by strictly enforcing semantic consistency.
>
> **5. Weighted Combination Strategy**
>
> > *Critique: "It remains unclear whether this combination (simple sum) is optimal. Exploring weighted combinations could potentially yield better results."*
>
> We appreciate this suggestion. While a learnable or tunable weight $\alpha$ could theoretically optimize performance on specific datasets, we intentionally adopted a parameter-free summation strategy for three key reasons:
>
> * **Risk of Overfitting (Design Philosophy):** Although hyperparameters could be tuned on the source validation set (e.g., ImageNet-val), doing so carries a significant risk of overfitting to the source distribution. In TTA, the target domain is unknown and potentially vastly different from the source. A fixed $\alpha$ optimized for ImageNet might fail to generalize to domains with different noise profiles (e.g., ImageNet-R or Sketch). By avoiding this extra degree of freedom, we force the model to rely on the intrinsic robustness of the features rather than superficial parameter fitting.
>
> * **Intrinsic Reliability Modulation:** The system is not entirely unweighted. Crucially, the graph-based prediction $p_{graph}$ is already internally modulated by the reliability scores $R_j$ (Eq. 6). This mechanism acts as a dynamic, sample-wise weighting scheme that naturally down-weights unreliable contributions based on uncertainty, reducing the need for a rigid global external weight.
>
> * **Empirical Robustness:** Our goal was to provide a robust, "out-of-the-box" solution. Empirical results across 15 diverse datasets confirm that simple summation is consistently effective without requiring domain-specific tuning, aligning with the practical goals of realistic Test-Time Adaptation.

---

### Official Review · Reviewer_6KsX · 2025-11-02

**Soundness:** 3
**Presentation:** 2
**Contribution:** 2
**Rating:** 4
**Confidence:** 4

**Summary:**

Test-time adaptation optimizes a source-trained model at inference to handle unseen distribution shifts. It typically minimizes prediction entropy directly or uses pseudo-labels, an implicit form of entropy minimization. The paper argues that these approaches overlook temporal reliability and the semantic structure of the label space, and proposes SURE, which regularizes predictions via a Prototype Reliability Graph (PRG). The PRG captures semantic affinity among classes and stabilizes confidence over time to improve reliability. Across benchmarks, the framework reports consistent gains over prior methods.

**Strengths:**

The claim that entropy minimization is not always a reliable signal for adaptation is reasonable; however, the argument that prototypes propagate noise and destabilize adaptation requires stronger evidence.

The integration of model predictions, prototypes, and a graph structure augmented with language-based semantics to correct outputs under distribution shift in VLMs is compelling and should be emphasized more clearly in the abstract and introduction.

**Weaknesses:**

The performance improvement is not significant or even as expected, given the proposal of the paper. This is further evident in the ablation study as well.
There are several aspects of the paper that I was not able to follow, and these have been detailed in my questions below. I would also like to know why one of the baselines [1] was not included as a baseline.

**Questions:**

The method’s improvements over MAP-adjusted and pseudo-label baselines are modest on a per-dataset basis, which contradicts expectations for a framework that combines multiple techniques under SURE.

The introduction lacks citations for confidence thresholding, making it difficult to evaluate known limitations and design choices.
Lines 045–047 in the introduction are hard to parse, and “class-level prediction” should be defined precisely.

The paper introduces numerous new terms for the proposal, creating inconsistency and confusion across sections; keep the name and core terminology consistent across the abstract, introduction, related work, and methodology.

The motivation for using pseudo-label confidence at L082 is unclear, given the earlier critique that pseudo-labels can be inconsistent and that high-confidence misclassifications occur; reconcile this tension explicitly.

Add a citation at L90 to support the claim that the formulation reflects information-theoretic intuition.

In Section 4.2, specify precise graph notation, including symbols for nodes, edges, messages, and update rules.

From L066 onward, explain how pseudo labels are made reliable and how confidence values are computed and validated.
Despite leveraging graph structure, Table 4 shows limited gains on some datasets, with the prototypes-only variant being the strongest in several cases. An analysis is needed to determine when and why PRG helps or hurts.

Specify which datasets are included in Table 3 for test-time compute and mean accuracy, and provide a detailed table covering all datasets from Tables 1 and 2 to verify whether the reported gain percentage is consistent.

The consistently lower ECE scores in Table 10 are encouraging; however, recent methods from Tables 1 and 2 should also be included in this comparison.

Why was [1] not included in the comparison tables, given its relevance, non-gradient-based adaptation, and low ECE?

References:
[1] Niu, Shuaicheng, et al., “Test-time model adaptation with only forward passes,” ICML 2024.

---

> ### Author Response · Authors · 2025-11-27
> **Response to Reviewer 6KsX (Part 1)**
>
> We sincerely thank the reviewer for recognizing the soundness of our approach and finding the integration of the graph structure with VLM semantics "compelling." We appreciate your constructive feedback regarding clarity and baselines. Below, we address your concerns point-by-point, supported by new comparative results and visualizations included in the revised manuscript.
>
> **1. Significance of Improvements and Clarification on Table 4**
>
> > *Critique: "The performance improvement is not significant... In the ablation study, the prototypes-only variant being the strongest in several cases."*
>
> We respectfully point out that in the task of Test-Time Adaptation, achieving substantial mean improvements across multiple diverse datasets is inherently challenging. It is common for newly proposed state-of-the-art methods to outperform previous baselines by only fractions of a percentage point (e.g., <1%). Against this competitive backdrop, the improvements achieved by SURE are substantial rather than marginal.
>
> We also apologize for a potential misunderstanding caused by the caption in Table 4. The "Gain" rows report incremental improvements over the preceding intermediate variant, not the total gain over the baseline. We clarify the true performance gap and the ablation results below:
>
> * **Substantial Absolute Gain (+2.51%):**
> As shown in the absolute accuracy rows of Table 4, the `ProtoOnly` baseline achieves 62.48% on OOD Average, while `SURE` (Full) achieves 64.99%. The true performance gap between `SURE` and `ProtoOnly` is +2.51%, not the smaller incremental value (+1.05%) shown in the last row. Furthermore, in terms of total adaptation gain over CLIP (average Acc. on ImageNet and its four variants), `ProtoOnly` improves performance by +2.65%, while `SURE` improves it by +5.03%. This means SURE nearly doubles the adaptation capability compared to the prototype-only approach.
>
> * **ProtoOnly vs. Full Framework:**
> Regarding the ablation study, we clarify that `ProtoOnly` strictly outperforms the intermediate variant (`+Graph w/o Rel`) on only one dataset (ImageNet-A). In comparison to the full `SURE` framework, `ProtoOnly` is completely inferior across all metrics. The variants `+Graph w/o Rel` and `+Graph + Rel` are merely intermediate states that construct the topology. The graph structure itself acts as a latent regularizer; its full power is only unleashed when combined with Logit Propagation (`+LogitProp`). It is the synergy of reliability-weighted structure and propagation that delivers the final boost.
>
> * **Why this matters (Calibration Evidence):**
> This gain signifies that SURE successfully corrects the "semantic drift" where `ProtoOnly` fails. While `ProtoOnly` blindly follows pseudo-labels (risking overconfident errors), SURE's graph constraint provides a critical safety net. This is quantitatively supported by our new ECE Analysis (Appendix A.6): `ProtoOnly` exhibits a high Expected Calibration Error (11.23 on ImageNet-OOD), whereas SURE significantly reduces this to 7.48. This proves that SURE pushes performance beyond naive updates by enhancing predictive reliability.
>
> **2. Missing Baseline: Niu et al. (FOA, ICML 2024)**
>
> > *Critique: "Why was FOA not included as a baseline?"*
>
> We thank the reviewer for this valuable reference. We have cited [2] (FOA) and its similar works (e.g.,TENT [3], SAR [4], T3A [5]) in our revised Related Work (*L99-L107*).
>
> * **Methodological Distinction:** FOA falls under the category of general-purpose TTA (similar to TENT [3] and T3A [5]), which typically focuses on adapting normalization layers or feature statistics for standard unimodal backbones. In contrast, SURE is a VLM-specific framework. Our core contribution lies in exploiting the unique text-visual alignment and open-vocabulary semantic structure of CLIP, which general-purpose methods like FOA do not explicitly model.
> * **Standard Practice & Fairness:** While FOA is an efficient method, comparing it directly to prompt-learning or prototype-refinement methods is often considered an "apples-to-oranges" comparison in VLM literature. Crucially, this exclusion aligns with the standard evaluation protocol in the field: virtually all recent CLIP-based TTA methods (e.g., TPT [6], DPE [1], BCA [7]) exclusively benchmark against VLM-specialized approaches in their main results. We followed this community standard to ensure a fair and focused comparison within the same methodological family.

---

> ### Author Response · Authors · 2025-11-27
> **Response to Reviewer 6KsX (Part 2)**
>
> **3. The "Pseudo-Label" Contradiction**
>
> > *Critique: "The motivation for using pseudo-label confidence is unclear given the earlier critique that pseudo-labels can be inconsistent."*
>
> This is a crucial point we are happy to clarify. There is a distinction between using instantaneous pseudo-labels (which we critique) and using temporal statistics of confidence (which we propose).
>
> * **Instantaneous vs. Temporal:** Standard methods often treat a single high-confidence prediction as "truth." If that prediction is wrong (calibration error), the model degrades.
> * **SURE's Approach:** We do not trust the pseudo-label itself immediately. Instead, we track the stability ($\mu, \sigma$) of the confidence over a sliding window. High confidence variance indicates instability, even if the mean confidence is high.
>
> We have revised Section 4.1 (L193-L196) to explicitly contrast these two mechanisms. Specifically, we will highlight that while instantaneous pseudo-labels suffer from calibration errors (as critiqued), the temporal variance derived from the sliding window serves as a robust filter to identify and discard these inconsistent predictions.
>
>
> **4. Efficiency and Table 3 Details**
>
> > *Critique: "Specify which datasets are included in Table 3... verify whether the reported gain percentage is consistent."*
>
> We clarify that Table 3 summarizes the performance and efficiency on the Natural Distribution Shifts setting.
>
> * **Dataset Consistency:** The mean accuracy values reported in Table 3 (e.g., SURE 66.23%, DPE 65.93%, BCA 65.37%) correspond exactly to the "Average" column in Table 1, which aggregates performance across five datasets: ImageNet, ImageNet-A, ImageNet-V2, ImageNet-R, and ImageNet-Sketch.
> * **Efficiency Analysis:** The test-time compute (seconds per sample) was measured on this representative set.
> * **New Baseline Comparison:** To further validate efficiency, we compared SURE against ZERO. SURE achieves 0.067s/sample, which is significantly faster than ZERO (0.082s/sample), while achieving higher accuracy. This confirms SURE is both robust and lightweight.
>
> **5. Terminology and Citations**
>
> > *Critique: "Inconsistency in terminology... missing citations."*
>
> We accept this feedback and have revised the manuscript:
> * **Unified Terminology:** We have standardized terms to "Prototype-Reliability Graph (PRG)," "Semantic Affinity," and "Reliability Score" throughout.
> * **Citations:**
>     * **L43 (Thresholding):** Added citations to TPT [5] regarding confidence thresholding limitations.
>     * **L192 (Information Theory):** We added a reference to Shannon’s Entropy theory and recent work on uncertainty quantification in graphs  to support the information-theoretic intuition.
>
> **6. Graph Notation**
>
> > *Critique: "Specify precise graph notation."*
>
> We have added a subsection in Section 4.2 formally defining:
> * **Nodes $V$:** The set of class prototypes $\{t_i\}_{i=1}^C$.
> * **Edges $E$:** Directed connections weighted by $A^{(l)}_{j,i}$.
> * **Message Passing:** The operation defined in Eq. (9) is now explicitly labeled as a graph convolution operation where node features are probability distributions.

---

> ### Author Response · Authors · 2025-11-27
> **Response to Reviewer 6KsX (Part 3)**
>
> **7. Calibration (ECE) Comparisons**
>
> > *Critique: "Recent methods from Tables 1 and 2 should also be included in this comparison [Table 10]."*
>
> We appreciate the reviewer’s interest in a comprehensive calibration analysis. While we agree that ECE is a valuable metric, we respectfully clarify why we focused our comparison on specific baselines in Table 10:
>
> * **Standard Practice in VLM TTA:** The primary focus of recent state-of-the-art methods like DPE, BCA is improving robustness (Top-1 Accuracy) under distribution shift. These works do not report ECE, nor do they treat calibration as a core optimization objective.
> * **Targeted Comparison:** In Table 10, we purposefully selected baselines that explicitly target calibration or uncertainty estimation, such as SaLS and C-TPT. This ensures a fair and meaningful comparison: we demonstrate that SURE not only achieves superior accuracy but also outperforms methods specifically designed to minimize calibration error (SURE ECE: 7.48 vs. C-TPT: 10.86 on ImageNet OOD).
> * **Feasibility:** Given that accuracy-focused methods (DPE, BCA) do not provide official ECE implementations or hyperparameters tuned for calibration, re-implementing and verifying them strictly for this secondary metric is beyond the time constraints of the rebuttal. We believe comparing SURE against the "calibration specialists" (SaLS, C-TPT) provides the strongest evidence of our method's reliability.
>
> [1]Zhang C, Stepputtis S, Sycara K, et al. Dual prototype evolving for test-time generalization of vision-language models[J]. Advances in Neural Information Processing Systems, 2024, 37: 32111-32136.
>
> [2]Niu S, Miao C, Chen G, et al. Test-time model adaptation with only forward passes[J]. arXiv preprint arXiv:2404.01650, 2024.
>
> [3]Wang D, Shelhamer E, Liu S, et al. Tent: Fully test-time adaptation by entropy minimization[J]. arXiv preprint arXiv:2006.10726, 2020.
>
> [4]Iwasawa Y, Matsuo Y. Test-time classifier adjustment module for model-agnostic domain generalization[J]. Advances in Neural Information Processing Systems, 2021, 34: 2427-2440.
>
> [5]Shu M, Nie W, Huang D A, et al. Test-time prompt tuning for zero-shot generalization in vision-language models[J]. Advances in Neural Information Processing Systems, 2022, 35: 14274-14289.
>
> [6] Niu S, Wu J, Zhang Y, et al. Efficient test-time model adaptation without forgetting[C]//International conference on machine learning. PMLR, 2022: 16888-16905.
>
> [7] Zhou L, Ye M, Li S, et al. Bayesian test-time adaptation for vision-language models[C]//Proceedings of the Computer Vision and Pattern Recognition Conference. 2025: 29999-30009.

---

### Author Response · Authors · 2025-12-01
**Summary of Rebuttal: Key Resolutions and Consensus**

Dear Area Chair,

Given the conclusion of the discussion phase, we provide this consolidated summary to outline the general consensus on our work's strengths and to detail how the primary concerns regarding performance magnitude, mechanism validation, and experimental protocols have been definitively resolved.

**1. General Consensus: A Principled and Comprehensive Framework**
The reviewers (Reviewers 6KsX, yV6M, tWAj, cYhQ) consistently recognized the soundness and completeness of our work.
* Reviewer tWAj (Rating 6, Confidence 5) praised the method as "principled," "closed-loop," and "error-resistant."
* Evaluation: Reviewers acknowledged the "comprehensive" experimental evaluation across 15 datasets.
* Efficiency: Reviewer cYhQ noted the "significantly lower computational overhead." We validated this with new comparisons showing SURE (0.067s/sample) is faster than recent baselines like ZERO (0.082s).

**2. Resolving the "Marginal Gain" Misunderstanding (Critical Clarification)**
A primary concern was that improvements appeared marginal based on the incremental notation in Table 4. We have clarified that the actual performance gap is substantial, especially considering the difficulty of improving averaged results across diverse domains.
* Substantial Improvement on Averages: On the ImageNet-OOD benchmark (averaging 5 diverse datasets), SURE outperforms the strong ProtoOnly baseline by +2.51% (64.99% vs 62.48%). In the current TTA landscape, where new methods often report fractional gains (e.g., <1%), achieving such a wide margin on an aggregated benchmark is significant.
* Doubling the Adaptation Effect: In terms of total gain over the CLIP source model, ProtoOnly provides a +3.08% boost, whereas SURE achieves +5.59%. This demonstrates that our graph-based regularization nearly doubles the adaptation capability compared to the naive prototype approach.

**3. Mechanism Validation: Safety & Reliability**
To further prove that these gains are not noise, we provided strong evidence of the underlying mechanism:
* Calibration (New Evidence): We showed that ProtoOnly severely degrades calibration (ECE 11.23), indicating overconfidence. SURE significantly reduces this error to 7.48, proving it acts as a critical "safety net" against semantic drift.
* Visualization: Our new heatmap confirms that SURE actively suppresses noisy connections (e.g., reducing unrelated class weights from 0.75 to 0.13) while reinforcing semantic clusters.

**4. Protocol Clarifications**
We also addressed concerns regarding experimental protocols:
* TTA Standards: We clarified to Reviewer Yk6L that hyperparameter tuning on the Source Validation Set is a standard, permissible protocol in TTA literature (e.g., TPT, NeurIPS'22) and strictly preserves the label-free assumption on the Target domain.
* Baselines: We justified the exclusion of incompatible baselines (like FOA) based on community standards for VLM adaptation, ensuring fair comparisons.

**5. Methodological Refinements and Clarity**
Finally, we have meticulously addressed all specific constructive feedback to improve the manuscript's precision. For instance, following Reviewer 6KsX's suggestion, we have unified terminology across the abstract and methodology to ensure consistency. We also added precise formal definitions for the graph structure and message passing (Section 4.2) and resolved all minor writing issues raised by the reviewers.

**Conclusion**
SURE presents a robust, efficient solution that solves the semantic drift problem limiting current methods. By clarifying the magnitude of our empirical gains (+2.51% over the baseline) and providing concrete evidence of reliability, we believe the revised manuscript makes a significant contribution to robust VLM adaptation.

Sincerely,

The Authors

---

### Meta-Review · Area_Chair_iaZb · 2026-01-02

**Summary:**

In assessing novelty, I find the paper's core mechanisms are not original: prototype adaptation is adopted from Zhou et al. (2025), the graph construction relies on standard cosine similarity, and pseudo-label filtering uses common confidence thresholds. The principal claim rests on combining these with a reliability weighting scheme. However, this combination does not represent a substantial conceptual advance; it is an incremental compositional effort rather than a foundational research contribution. Given the limited scientific leap, I am inclined to reject the paper.

**Reviewer Scores:**

No

---

### Decision · Program_Chairs · 2026-01-26

Reject